# Essential role of the *Crk* family-dosage in DiGeorge-like anomaly and metabolic homeostasis

Akira Imamoto[1] , Sewon Ki[2], Leiming Li[1], Kazunari Iwamoto[3], Venkat Maruthamuthu[4], John Devany[5] , Ocean Lu[1] , Tomomi Kanazawa[3], Suxiang Zhang[3], Takuji Yamada[6], Akiyoshi Hirayama[7], Shinji Fukuda[7,8,9,10], Yutaka Suzuki[11], Mariko Okada[2,3,12] 

***CRK*** and ***CRKL*** (***CRK-like***) encode adapter proteins with similar biochemical properties. Here, we show that a 50% reduction of the family-combined dosage generates developmental defects, including aspects of DiGeorge/del22q11 syndrome in mice. Like the mouse homologs of two 22q11.21 genes *CRKL* and *TBX1*, *Crk* and *Tbx1* also genetically interact, thus suggesting that pathways shared by the three genes participate in organogenesis affected in the syndrome. We also show that *Crk* and *Crkl* are required during mesoderm development, and *Crk/Crkl* deficiency results in small cell size and abnormal mesenchyme behavior in primary embryonic fibroblasts. Our systems-wide analyses reveal impaired glycolysis, associated with low Hif1a protein levels as well as reduced histone H3K27 acetylation in several key glycolysis genes. Furthermore, *Crk/Crkl* deficiency sensitizes MEFs to 2-deoxy-D-glucose, a competitive inhibitor of glycolysis, to induce cell blebbing. Activated Rapgef1, a Crk/Crkl-downstream effector, rescues several aspects of the cell phenotype, including proliferation, cell size, focal adhesions, and phosphorylation of p70 S6k1 and ribosomal protein S6. Our investigations demonstrate that Crk/Crkl-shared pathways orchestrate metabolic homeostasis and cell behavior through widespread epigenetic controls.

## Introduction

*CRK* and *CRKL* (*CRK-like*), two paralogs of the *CRK* gene family, are localized to 17p13.3 and 22q11.21 in the human genome, respectively. *CRK* was first identified as the avian oncogene *v-CRK*, followed by the discovery of its cellular counterpart. *CRKL* was later identified in human chromosome 22q11 based on its sequence similarities to *CRK* (Feller, 2001; Birge et al, 2009). Evolutionary evidence suggests that the two genes were generated by chromosomal duplication in the common vertebrate ancestor (Shigeno-Nakazawa et al, 2016). Despite their possible redundancy, *CRKL* has been implicated in DiGeorge syndrome (DGS) as a dosage-sensitive gene that also shows genetic interactions with *TBX1*, a key 22q11.21 gene (Guris et al, 2006; Racedo et al, 2015), whereas ~90% of DGS patients have a heterozygous 3-Mb microdeletion at 22q1.21, including these two and several other genes (McDonald-McGinn et al, 2015).

Although haploinsufficiency of *TBX1* has been strongly implicated in DGS, deficiency of mouse *Crkl* alone also affects normal development of anterior/frontal structures, including facial features, great arteries, heart, thymus, and parathyroid, as well as posterior structures, including genitourinary (GU) tissues, as collectively manifested as a condition that resembles DiGeorge anomaly (Guris et al, 2001; Racedo et al, 2015; Haller et al, 2017; Lopez-Rivera et al, 2017). *CRKL* point mutations have also been identified among a large cohort of patients with renal agenesis or hypodysplasia (Lopez-Rivera et al, 2017). A distal region of the common deletion that includes *CRKL* has been linked to GU defects among 22q11.2DS patients, and haploinsufficiency of *Crkl* results in abnormal GU development in mice (Haller et al, 2017; Lopez-Rivera et al, 2017). Although *CRKL* coding mutations have not been linked to DGS without a 22q11 deletion, a recent study has identified non-coding mutations predicted to affect *CRKL* expression in the hemizygous region of the common 22q11 deletion with conotruncal defects (Zhao et al, 2020). Therefore, a reduction of *CRKL* expression below 50% may contribute to expressivity and penetrance known to be highly variable in DGS. On the other hand, *CRK* has not been established with a firm link to congenital disorders to date, although it is localized to the chromosomal region associated with Miller–Dieker syndrome

---

[1]The Ben May Department for Cancer Research, The University of Chicago, Chicago, IL, USA   [2]RIKEN Integrative Medical Sciences, Tsurumi, Yokohama, Kanagawa, Japan   [3]Institute for Protein Research, Osaka University, Suita, Osaka, Japan   [4]Department of Mechanical and Aerospace Engineering, Old Dominion University, Norfolk, VA, USA   [5]Department of Physics, The University of Chicago, Chicago, IL, USA   [6]Department of Life Science and Technology, Tokyo Institute of Technology, Meguro, Tokyo, Japan   [7]Institute for Advanced Biosciences, Keio University, Tsuruoka, Yamagata, Japan   [8]Intestinal Microbiota Project, Kanagawa Institute of Industrial Science and Technology, Kawasaki, Kanagawa, Japan   [9]Transborder Medical Research Center, University of Tsukuba, Tsukuba, Ibaraki, Japan   [10]PRESTO, Japan Science and Technology Agency, Kawaguchi, Saitama, Japan   [11]Department of Computational Biology and Medical Sciences, Graduate School of Frontier Sciences, The University of Tokyo, Kashiwa, Chiba, Japan   [12]Center for Drug Design and Research, National Institutes of Biomedical Innovation, Health and Nutrition, Ibaraki, Osaka, Japan

Correspondence: aimamoto@uchicago.edu; mokada@protein.osaka-u.ac.jp
Leiming Li's present address is AbbVie, North Chicago, IL, USA

---

(Bruno et al, 2010). Nevertheless, mouse phenotypes from genetic ablations of either *Crkl* or *Crk* indicate that neither *Crk* nor *Crkl* alone is sufficient for normal development (Guris et al, 2001; Park et al, 2006).

*CRK* and *CRKL* encode adapter proteins, consisting of SRC homology 2 and 3 domains (SH2 and SH3, respectively) without known catalytic activities in an SH2-SH3-SH3 configuration, whereas alternative splicing generates CRK "isoform b" (commonly noted as "CRK-I" in contrast to the full length "isoform a" as "CRK-II") that does not include the C-terminal SH3 domain (Feller, 2001; Birge et al, 2009). Most CRK/CRKL SH2-binding proteins have been identified as transmembrane proteins (such as growth-factor/cytokine receptors and integrins) and their cytosolic components (Feller, 2001; Birge et al, 2009). The task of inferring the specifics of their biological functions has been challenging due partly to co-expression of CRK and CRKL. Several broadly expressed SH3-binding proteins such as RAPGEF1 (C3G), DOCK1 (DOCK180), and ABL also co-exist in a single cell in which they engage with multiple input signals to elicit context-dependent coordinated responses.

To address the challenges noted above, we have used mouse models in which either or both *Crk* and *Crkl* can be disrupted conditionally. Developmental defects in the mouse models have similarities to DGS, and normal development of the affected tissues is sensitive to the combined gene dosage of the *Crk* and *Crkl* genes. Furthermore, we report here a dosage-sensitive interaction between *Crk* and *Tbx1*, similar to the genetic interaction we previously reported between the mouse homologs of two 22q11.21 genes, *CRKL* and *TBX1* (Guris et al, 2006). Therefore, investigation of the pathways at the functional/genetic intersection of *Crk* and *Crkl* will be important for elucidating the mechanisms that underlie DiGeorge and other related congenital syndromes. As we have found that the mesoderm requires *Crk* and *Crkl*, we have chosen primary MEFs as a mesoderm model. A series of unbiased systems-level analyses and functional validations have revealed the shared dosage-sensitive roles of *Crk* and *Crkl* in coordinating glucose metabolism and cell size homeostasis by integrating regulatory pathways partly through widespread epigenetic modifications.

# Results

## Deficiency of Crk, the paralog of Crkl, targets the heart and arch-derived tissues

To probe the functional significance of the *Crk* family members, we targeted the mouse *Crk* gene with a conditional approach by inserting *loxP* sites upstream and downstream of Exon 1 (*Crk^f* allele; Fig S1). A germ-line *Crk* null allele (*Crk^d* allele) was generated by Cre-mediated recombination in the epiblast using a *Meox2* Cre knock-in strain (Tallquist & Soriano, 2000), followed by backcrosses with wild-type C57BL/6 mice to segregate out *Meox2^Cre*. In addition to the developmental defects previously reported in another *Crk*-deficient mutant (Park et al, 2006), we noted that homozygous *Crk^{d/d}* embryos displayed some aspects reminiscent of DiGeorge anomaly despite the fact that *CRK* is not a 22q11 gene in humans (Figs 1A–D and 1A'–D'). Among three *Crk^{d/d}* embryos histologically examined, all three cases displayed

ventricular septal defects (VSD) (Fig 1D), whereas one case accompanied an interrupted arch of aorta (IAA-B, Fig 1D), another case a right aortic arch, one case a d-transposition of the great arteries (Fig 1D), two cases with a double-outlet right ventricle (Fig 1D), two cases with a cleft palate (Fig 1A), and two cases with cervical thymic lobes outside of the thoracic cavity (Fig 1B).

## Compound heterozygosity of Crk and Crkl is sufficient to generate an embryonic phenotype

*Crk* and *Crkl* were expressed in largely overlapping patterns at E10.5, and the *Crk*-deficient phenotype was similar to that of *Crkl* (Figs S2 and S3) (Guris et al, 2001). Therefore, we hypothesized that their phenotypes may be attributed to a dosage-sensitive reduction in their common functions. In addition to the *Crk* conditional allele, we used a mouse strain that we previously generated with a conditional mutation in the *Crkl* gene in which exon 2 is flanked by two *loxP* sites as *Crkl^{f2}* allele (Haller et al, 2017; Lopez-Rivera et al, 2017). We first confirmed that the *Crkl*-deficient embryonic phenotype generated by *Crkl^{f2/f2}* and *Meox2^{Cre/+}* strains recapitulated the *Crkl* null phenotype generated by deletion of *Crkl* exon 1, including arch artery and thymic defects (Fig S3). As predicted, compound heterozygotes for *Crk^f* and *Crkl^{f2}* with *Meox2^Cre* exhibited an embryonic phenotype at E16.5, including severe edema and enlarged blood vessels, a cleft palate, IAA-B, and right-sided aortic arch accompanied by ventricular septal defect and small thymic lobes (Fig 1E–H). IAA-B was reproducibly observed in *Crk/Crkl* compound heterozygous embryos (Fig 1I). This phenotype was similar in multiple aspects to the phenotypes from homozygous deficiency of either *Crk* or *Crkl* (Figs 1A–D and S3). Furthermore, compound heterozygotes between *Crk* and *Tbx1* showed embryonic phenotypes at E16.5 with greater penetrance and expressivity than that of either *Crk* or *Tbx1* single heterozygotes (Table S1). As these phenotypes shared a constellation of DGS-like defects, our observations raise the hypothesis that DiGeorge and related syndrome may result from genetic and environmental assaults on a part of the network sensitive to and commonly dependent on the *CRK* family genes as well as *TBX1*.

## The mesoderm requires at least two copies of the Crk family-combined gene dosage

To further investigate shared roles that *Crk* and *Crkl* may play in development, we generated *Crk* and *Crkl* deficiency in the mesoderm using *Mesp1^Cre* (Saga et al, 1999). Some mice survived 50% family-combined gene dosages reduced in the mesoderm lineages without an overt phenotype in three genotypes: *Crk^{f/f};Mesp1^{Cre/+}*, *Crkl^{f2/f2};Mesp1^{Cre/+}*, and *Crk^{f/+};Crkl^{f2/+};Mesp1^{Cre/+}*. However, further dosage reduction leaving only one copy of either *Crkl* or *Crk* in the mesoderm (*Crk^{f/f};Crkl^{f2/+};Mesp1^{Cre/+}*, and *Crk^{f/+};Crkl^{f2/f2};Mesp1^{Cre/+}*, respectively) resulted in abnormal embryos, associated with an enlarged heart that failed to undergo looping when examined at E9.5 (Fig 1J and K). In addition, they also had smaller numbers of somites with a large proportion of the paraxial mesoderm left unsegmented compared with that of control embryos. Although vasculogenesis was initiated in the yolk sac mesoderm, the vascular plexus failed to undergo remodeling in *Crk^{f/+};Crkl^{f2/f2};Mesp1^{Cre/+}* embryos recovered at E9.5 (Fig 1M). It is also noteworthy that

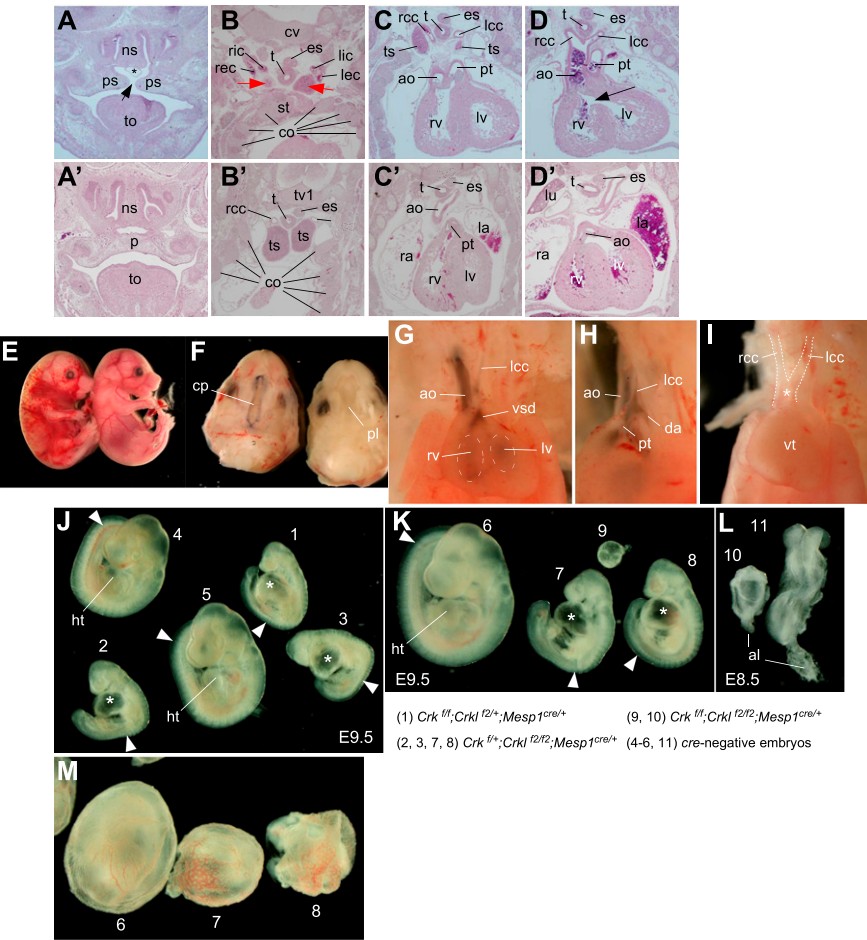

**Figure 1. Embryonic phenotypes from deficiencies of *Crk* and *Crkl* in mice.**

**(A, B, C, D)** Histologic sections from an E16.5 embryo lacking *Crk* (*Crk*$^{d/d}$) showed defects, including a cleft palate (arrow in panel A), cervical/extra-thoracic thymic lobes (red arrows in panel B, ts in panel C), d-transposition of aorta and pulmonary trunk associated with double-outlet right ventricle (C) and ventricular septal defect (arrow in panel D). We also noted a condition known as an interrupted aortic arch type B (IAA-B) in panel (D) and other sections (not shown). Asterisk in panel (A) indicates a dilated blood vessel. Accompanied panels (A′, B′, C′, and D′) show sections from a wild-type littermate corresponding to sections (A, B, C, and D), respectively. Abbreviations used in the panels are as follows: ns, nasal septum; ps, palatal shelf; to, tongue; rcc, right common carotid artery; lcc, left common carotid artery; ric, right internal carotid artery; lic, left internal carotid artery; rec, right external carotid artery; lec, left external carotid artery; cv, cervical vertebra; tv1, thoracic vertebra 1; tv2, thoracic vertebra 2; es, esophagus; t, trachea; ts, thymus; st, top of the sternum (manubrium); co, ribs (costae); ao, aorta; pt, pulmonary trunk; rv, right ventricle; and lv, left ventricle. **(E, F, G, H, I)** Compound heterozygosity for *Crk* and *Crkl* deficiency resulted in an embryonic phenotype at E16.5. Timed mating was set up between *Meox2*$^{cre}$/+ and *Crk*$^{f/f}$; *Crkl*$^{f2/f2}$ parents to drive cre-dependent recombination in the epiblast. Compound heterozygotes (*Crk*$^{f/+}$; *Crkl*$^{f2/+}$;*Meox2*$^{cre/+}$) showed severe edema and subcutaneous hemorrhage at E16.5 (left, E), associated with a cleft palate (F) and abnormal great arteries and heart (G, H). Ink injection into the right ventricle revealed an abnormal pattern of the great arteries such as enlarged aorta without forming a left-sided arch of aorta (ao, G) as well as a ventricular septal defect (G), as ink flowed into the left ventricles from the right ventricle (dotted ellipses, G). When viewed from the left side (panel H), pulmonary trunk abnormally branched into the left common carotid artery via the ductus arteriosus connected to the descending aorta. A similar case of interrupted arch of aorta type B was found in another compound heterozygote in the same litter (I). Asterisk indicates an abnormal outflow tract externally suspected to be a persistent truncus arteriosus. The compound heterozygote also exhibited a small cervical thymic lobe, which was removed before examination of the great arteries. cp, cleft palate; pl, palate (closed); rcc, right common carotid artery; lcc, left common carotid artery; ao, aorta; rv, right ventricle; lv, left ventricle; vsd, ventricular septal defect; da, descending aorta. **(J, K, L, M)** Early developmental defects were observed in E8.5 and E9.5 mouse embryos when combined *Crk* and *Crkl* deficiency was induced in the mesoderm driven by *Mesp1*$^{cre}$. The genotypes of the individual embryos shown (numbered from 1 through 11) are indicated below the panels (K, L). Panels (J, K) show lateral views of embryos isolated at E9.5. Panel (L) shows dorsal views of two E8.5 embryos. Note that *Crk*$^{f/f}$;*Crkl*$^{f2/+}$;*Mesp1*$^{cre/+}$ and *Crk*$^{f/+}$;*Crkl*$^{f2/f2}$;*Mesp1*$^{cre/+}$ embryos were phenotypically similar (embryo 1 compared with embryos 2, 3, 7, and 8). Asterisks indicate enlarged hearts without proper looping and chamber development. Arrowheads indicate the position of the posterior most somite visually identifiable, thereby indicating a delay in somitogenesis in *Crk*$^{f/f}$;*Crkl*$^{f2/+}$;*Mesp1*$^{cre/+}$ and *Crk*$^{f/+}$;*Crkl*$^{f2/f2}$;*Mesp1*$^{cre/+}$ embryos compared with cre-negative control embryos. ht, heart; al, allantois. Panel (M) shows embryos 6, 7, and 8 in yolk sac. Note a delay in vascular remodeling in embryos 7 and 8, compared with the cre-negative control embryo (embryo 6).

(1) *Crk* $^{f/f}$;*Crkl* $^{f2/+}$;*Mesp1*$^{cre/+}$ (9, 10) *Crk* $^{f/f}$;*Crkl* $^{f2/f2}$;*Mesp1*$^{cre/+}$

(2, 3, 7, 8) *Crk* $^{f/+}$;*Crkl* $^{f2/f2}$;*Mesp1*$^{cre/+}$ (4–6, 11) *cre*-negative embryos

embryos with only one copy of either *Crkl* or *Crk* (*Crk* $^{f/f}$;*Crkl*$^{f2/+}$; *Mesp1*$^{Cre/+}$ or *Crk* $^{f/+}$;*Crkl* $^{f2/f2}$;*Mesp1*$^{Cre/+}$, respectively) showed similar morphological defects. These results indicate that a developmental threshold requires at least 50% of the family-combined gene dosage during heart and somite development as well as in the yolk sac mesoderm, through their shared functions. We also identified *Crk*/*Crkl* double-deficient embryos (*Crk* $^{f/f}$;*Crkl* $^{f2/f2}$; *Mesp1*$^{Cre/+}$) in the genetic crosses. They were much smaller than either *Crk* $^{f/f}$;*Crkl* $^{f2/+}$;*Mesp1*$^{Cre/+}$ or *Crk* $^{f/+}$;*Crkl* $^{f2/f2}$;*Mesp1*$^{Cre/+}$ embryos, at E9.5 (Fig 1K). When isolated at E8.5, double-deficient embryos resembled the size and appearance of E7.5 embryos (Fig 1L). Because normal onset of gastrulation is marked by the emergence of *Mesp1*-positive mesoderm starting around E6.5 in mice (Saga et al, 1999), these results indicate that *Crk* and *Crkl* are absolutely required immediately after mesoderm induction.

## Morphological and behavioral phenotypes in primary MEFs

The results above suggest that mesodermal cells may provide a useful system to investigate the shared functions of *Crk* and *Crkl*. To this end, we isolated primary MEFs at E11.5 as a model for mesodermal cells. Using *Rosa26-creERT2* in the background of the *Crk* $^{f/f}$, *Crkl* $^{f2/f2}$, or *Crk* $^{f/f}$;*Crkl* $^{f2/f2}$ genotypes, deficiency of either or both *Crk* and *Crkl* were induced by 4-hydroxytamoxifen (4OHT) over a course of 72 h in *Crk* $^{f/f}$;*Crkl* $^{f2/f2}$;*R26*$^{creERT2/+}$ MEFs (Fig 2A) (Ventura et al, 2007). *Crk*-deficient MEFs did not show abnormal motility when plated in a clonal cell density (Fig S4). However, we noted that the pH indicator phenol red in the culture medium did not turn yellow when the cells were in a high density, associated with a morphology change (Fig S4). These results demonstrated that *Crk* plays a role in collective morphology, whereas the medium pH implicated an

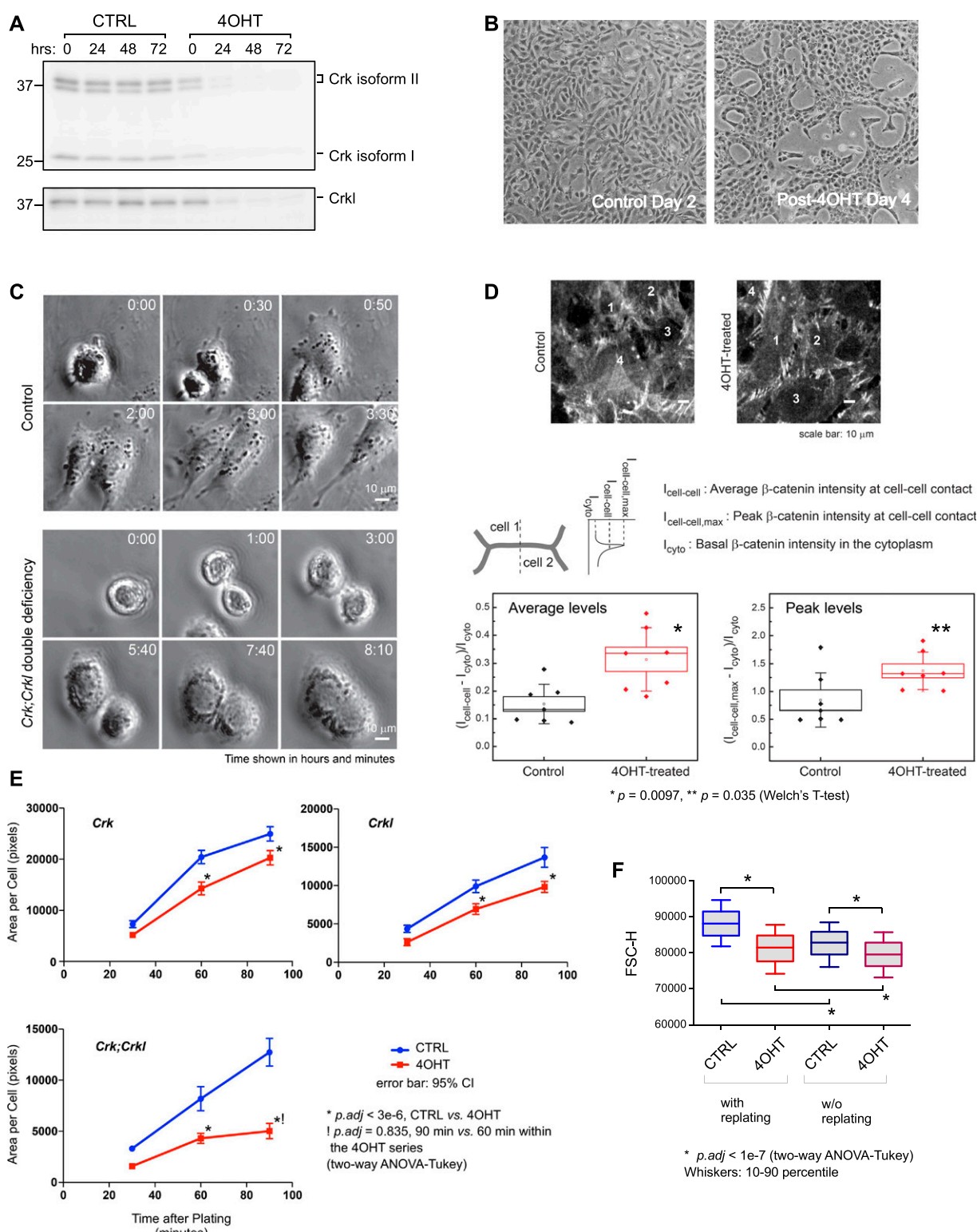

**Figure 2. *Crk;Crkl* deficiency induces multiple defects in MEFs.**
**(A)** Crk and Crkl protein levels were determined in time course in immunoblots upon induction of *Crk;Crkl* deficiency. *Crk f/f;Crkl f2/f2;R26creERT2/+* MEFs were induced for *Crk;Crkl* deficiency by 4-hydroxytamoxifen (4OHT) treatment for 24 h, whereas a control group was treated with vehicle only (CTRL). Cell lysates were isolated from MEFs at each time point indicated above each lane (0 h was the time right before 4OHT/vehicle addition). **(B)** *Crk;Crkl* double deficiency results in slow cell growth and altered population morphology. Pictures were taken posttreatment day 2 for control and day 4 for *Crk;Crkl* deficiency group. Note that the cell density of *Crk;Crkl*-deficient MEFs on day 4 is similar to that of control group on day 2, and that population morphology is distinguishable between the 4OHT and control groups. **(C)** Time lapse images show cell division from a single MEF (identified at 0:00 time) to two daughter cells in each group. In the control, cells migrated away from each other after division and were no

impaired metabolic state. Upon induction of *Crk*/*Crkl* double deficiency, the cell morphology changed more drastically from a typical fibroblastic appearance to a compact/condensed appearance (Fig 2B). To explore the basis of this collective morphology, we took time-lapse videos of dividing cells (Fig 2C). Normally, fibroblast-like cells show repulsive movements upon cell–cell contacts, known as contact inhibition of locomotion (CIL) (Roycroft & Mayor, 2016). Likewise, two daughter cells moved apart in the control group upon cell division (Fig 2C). In contrast, the daughter cells in the *Crk*/*Crkl*-deficient group did not separate despite their cell–cell contacts, thus exhibiting a failure related to CIL (Fig 2C). The cell junctional marker β-catenin showed a greater accumulation to cell–cell junctions in the deficiency-induced group than control, thus indicating elevated cell–cell adhesion in the deficiency-induced MEFs (Fig 2D). A failure of post-mitotic CIL and increased cell–cell contacts may explain the abnormal population morphology in Fig 2B, thus demonstrating that *Crk* and *Crkl* play a pivotal role in cell–cell haptic communication and behavior.

### Essential roles of Crk and Crkl in spreading and cell size

We next determined the effects of *Crk* and *Crkl* deficiency, individually or combined, on cell spreading (Fig 2E). In our modified spreading assay, we measured the surface area that each attached cell occupied over time on gelatin-coated plates rather than counting the number of spreading cells at each time point. We found that compared with control MEFs, the process of spreading was slower in deficiency of *Crk* or *Crkl* individually, and further reduced in *Crk*/*Crkl* double deficiency as indicated by the slope of spreading curves (Fig 2E).

During tissue culture, we became aware that the same number of *Crk*/*Crkl* double-deficient primary MEFs made visibly smaller pellets than that of control MEFs when harvested by dissociation and centrifugation. The observation suggested the possibility that individual *Crk*/*Crkl* double-deficient MEFs may be smaller than that of control cells. Normally, cells undergo a controlled cell-size oscillation during cell cycle to maintain their sizes in a population (Lloyd, 2013; Ginzberg et al, 2015). Therefore, we estimated the size of primary MEFs in the G1 phase by light scatter measurements in FACS analysis (Fig 2F). As anticipated, induction of *Crk*/*Crkl* double deficiency resulted in a size distribution shift smaller than that of control primary MEFs, whereas *Crk*/*Crkl* double-deficient MEFs cells appeared to stay in the G1 phase for a longer time than the control group (Fig S5). We also noted that the cell size was

smaller when kept confluent for 4 d, compared with the groups that were split on Day 2 to avoid overcrowding (all groups received daily media change). Therefore, these results demonstrate that *Crk* and *Crkl* are essential for cell size homeostasis in G1, whereas additional cell density–dependent mechanism may also operate in parallel.

### Transcriptome pathways dependent on Crk and Crkl

The complex phenotypes in development and in MEFs suggested involvement of *Crk* and *Crkl* in multiple pathways. To gain insight into the impaired network from a vantage view point, we conducted a systems-level analysis by RNA-Seq in the primary MEFs in which deficiency of each or both *Crk* and *Crkl* can be induced by 4OHT (Fig 3). Differential expression (DE) was determined between deficiency-induced and uninduced groups of primary MEFs in pair per single embryo, using four independent embryos for each genotype with an false discovery rate (FDR) cutoff of *p.adj* < 0.05 (Table S2 and Fig S6). Fig 3A shows a heat map of the DE genes in protein synthesis ("EIF2 Signaling," "Regulation of EIF4 and p70 S6K Signaling," and "mTOR Signaling"), growth factor signaling ("VEGF Signaling," "IGF-1 Signaling," "PTEN Signaling"), adhesion and cytoskeletal signaling ("Integrin Signaling," "Actin Cytoskeleton Signaling," "FAK Signaling," "Paxillin Signaling," "Signaling by Rho GTPases," "RhoA Signaling," "Ephrin Receptor Signaling," "Ephrin A Signaling," "Gap Junction Signaling") (Supplemental Data 1).

Upon conducting "set operations," we identified ~400 genes in the common intersection among either *Crk* or *Crkl* single deficiency and *Crk*/*Crkl* combined deficiency (Fig 3B, subset "red"; Supplemental Data 2). The DE genes in this subset are likely regulated by the pathways that *Crk* and *Crkl* share in a "family dosage-sensitive" manner. In addition to subset "red," deficiency of either *Crk* or *Crkl* also resulted in DE in subsets "*orange*" and "*yellow*," respectively (Fig 3B). While the DE genes identified in subsets "red," "*orange*," and "*yellow*" were sensitive to a single deficiency of either *Crk* or *Crkl*, subset "*green*" represents genes dependent on the shared pathways that combined deficiency of both *Crk* and *Crkl* was needed to disrupt (Fig 3B). In other words, either *Crk* or *Crkl* was sufficient to maintain normal expression of the genes in subset "*green*" in primary MEFs. Therefore, subset "*green*" may represent genes for which *Crk* and *Crkl* may be redundant. Although we identified DE genes in *Crk* or *Crkl* deficiency not observed in *Crk*/*Crkl* double-deficient MEFs, numbers of these DE genes were too small to draw interpretations in the current study.

---

longer visible together within the field after 4 h. In contrast, *Crk;Crkl*-deficient cells (4OHT-treated) stayed attached with one another, forming a two-cell island after division. Movies are available as supplemental materials. **(D)** Cell–cell contacts were analyzed by immunostaining with anti-Ctnnb1 (β-catenin) antibody. Individual cells are labeled by numbers. β-catenin localization highlights defined zipper-like cell–cell contacts in *Crk;Crkl*-deficient MEFs (4OHT-treated) compared with control MEFs. Box and whisker plots show quantitative comparisons of β-catenin levels across cell–cell contacts shown in the images above the plots. Diamonds indicate intensity values derived from tracing seven separate regions that encompass cell–cell contacts. Two plots demonstrate that *Crk;Crkl*-deficient MEFs (4OHT-treated) had greater accumulation of β-catenin at cell–cell contacts compared with control as shown in both average and maximum levels of the β-catenin staining ("average levels" and "peak levels," respectively). **(E)** The area of spreading was measured for individual cells after replating on a gelatin-coated surface (72 cells in each group). Note that whereas MEFs lacking either *Crk* or *Crkl* showed smaller spread area at 60 and 90 min, *Crk;Crkl* double deficiency induced greater degrees of spreading defects over time and did not show a significant increase in the spread area between 60 and 90 min, thus suggesting that cell spreading reached a lower plateau compared with their control (and that of either *Crk* or *Crkl* deficiency). **(F)** The cell size was estimated in dissociated *Crk;Crkl*-deficient MEFs and their control in the G1 phase (4OHT and CTRL, respectively). FSC-H values (forward scatter height) were analyzed and illustrated in a box-and-whisker plot. See Fig S5 for propidium iodide–binding profiles and gating information for the FACS analysis. Each treatment group was subdivided into two subgroups with or without a split on day 2 posttreatment to adjust and maintain low cell density until harvest on day 3 posttreatment.

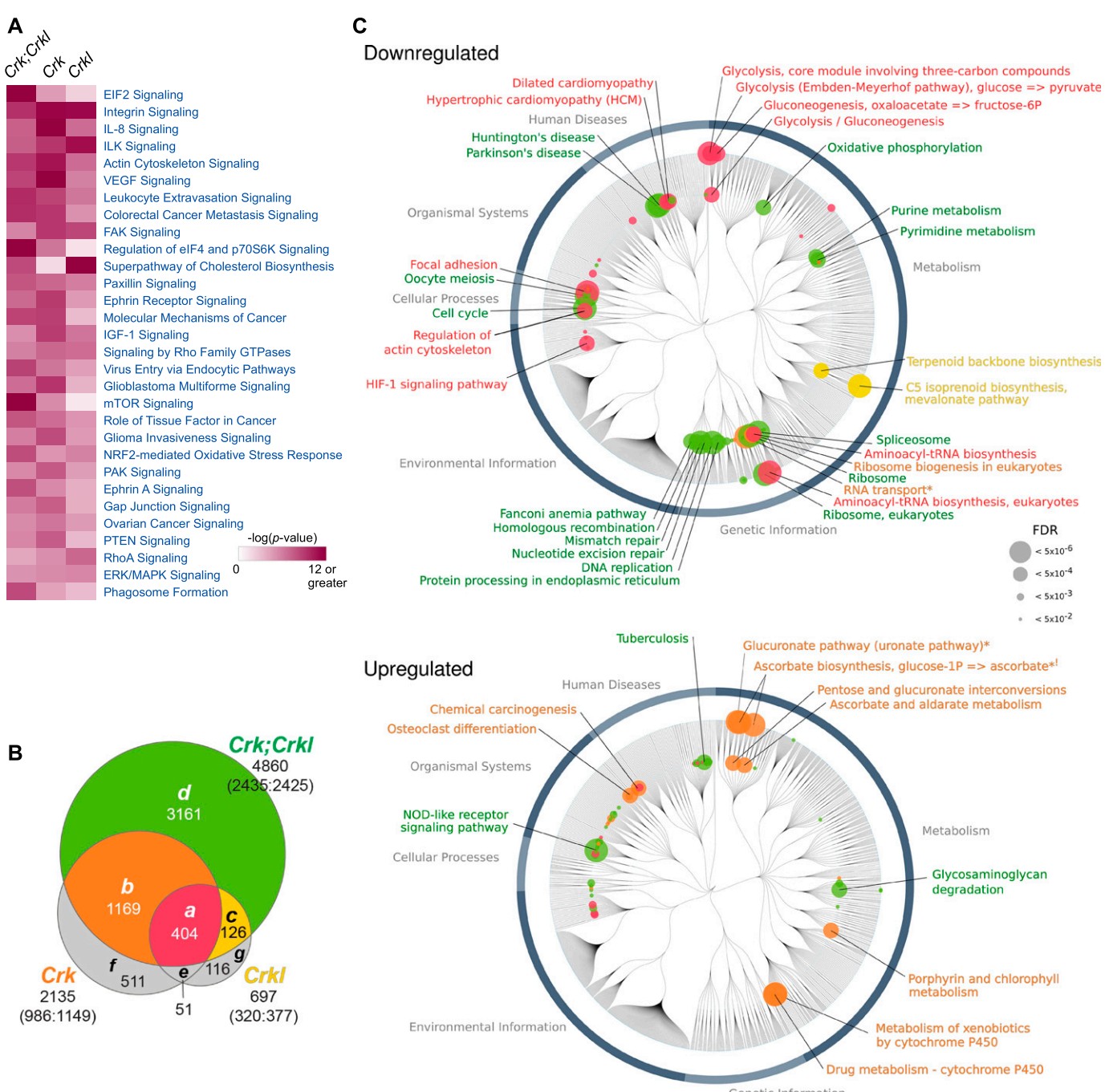

**Figure 3. Crk and Crkl deficiencies affect numerous pathways.**
**(A)** The heat map shows a list of the top 30 pathways based on comparison analysis of the *Crk* and/or *Crkl* single and double deficiency groups in Ingenuity Pathway Analysis (QIAGEN; see Supplemental Data 1). RNA-Seq experiments were performed on RNA isolated from four independent primary MEF populations for each genotype as described in the Materials and Methods section as well as in Table S2 and Supplemental Data 1. Differentially expressed genes (DE genes) were identified by Benjamini–Hochberg adjusted *P*-values (*p.adj*) smaller than 0.05 using DESeq2. **(B)** Deficiencies for *Crk* and *Crkl* genes, separately or combined, resulted in overlapping lists of DE genes, categorized into subsets *a–g* as shown in the Venn's diagram. Subsets *a*, *b*, *c*, and *d* are referred to subsets "red," "orange," "yellow," and "green" hereafter (Supplemental Data 2). The number in each subset indicates the number of DE genes in the subset. The number below the gene symbol (*Crk*, *Crkl*, or *Crk;Crkl*) indicates the total number of DE genes identified in the gene deficiency. The numbers in parentheses separated by colon show the numbers of genes up-regulated versus down-regulated. Note that deficiency of either *Crk* or *Crkl* was sufficient to disrupt normal expression of the genes in subset "red," whereas the genes in subset "green" tolerated single gene disruption of either *Crk* or *Crkl*. **(C)** The DE genes in subsets "red," "orange," "yellow," and "green" were analyzed for their enrichment into pathways using KEGG (Kyoto Encyclopedia of Genes and Genomes). The node circles and annotations are color-coded as appeared in panel (B). Nodes are labeled only for the KEGG modules and pathways with Storey's *q*-values smaller than 0.0005 (shown as FDRs), whereas node circles are shown for the pathways/modules with a *q*-value < 0.05. The diameter of node circle is proportional to $-\log_{10}(q\text{-value})$.

## KEGG analysis

To further explore the dysregulated pathways, we categorized the DE genes either "*down-regulated*" or "*up-regulated*" in each subset (Supplemental Data 2). In subset "red," we noted that *down-regulated* DE genes were enriched in several KEGG "pathways" and "modules," including glycolysis, aminoacyl-tRNA biosynthesis, HIF-1 signaling, regulation of actin cytoskeleton, and focal adhesion (Fig 3C, red circles). On the other hand, *up-regulated* DE genes in subset "red" did not show significant enrichment in a KEGG pathway or module. *Down-regulated* DE genes in subset "*orange*" were associated with ribosome biogenesis and RNA transport, whereas the *up-regulated* genes were mapped to the glucuronate pathway and cytochrome P450-mediated drug metabolism (Fig 3C, orange circles). *Down-regulated* genes in subset "*yellow*" were enriched in C5-isoprenoid biosynthesis/mevalonate pathway, suggesting a specific role for Crkl in biosynthesis of cholesterol and other isoprenoids, whereas no enrichment was identified in the pathways or modules for *up-regulated* genes (Fig 3C, yellow circles). Subset "*green*" included many DE genes enriched in *down-regulated* pathways, including oxidative phosphorylation, purine/pyrimidine metabolism, spliceosome, ribosome, DNA repair and replication, and cell cycle (Fig 3C, green circles). A few pathways, of which most noticeable was NOD-like receptor signaling, appeared to be *up-regulated* in subset "green," thus implicating a redundancy between *Crk* and *Crkl* in regulating inflammasomes (Strowig et al, 2012; Wen et al, 2013).

## Validating the role of Crk and Crkl in glycolysis

The transcriptome analysis above implicated shared family-critical roles for *Crk* and *Crkl* in glycolysis and other metabolic pathways (Fig 3). Using capillary electrophoresis time-of-flight mass spectrometry (CE-TOFMS), we found that several metabolites in the central glucose metabolism pathway were decreased (Fig 4A, squares filled with shades of blue; Supplemental Data 3), consistent with reduced transcript levels of several genes encoding glycolysis enzymes along the same pathway (Fig 4A, small circles filled with shades of blue). Several metabolites and glycolytic enzymes were affected not only in *Crk/Crkl*-double deficiency but also in MEFs deficient for either *Crk* or *Crkl* (Fig 4A, squares and circles enclosed by magenta-colored line). Reduced mRNA levels of several glycolysis enzymes initially identified by RNA-Seq were validated by quantitative real-time RT-PCR (Fig 4B). Furthermore, chromatin immunoprecipitation (ChIP) followed by quantitative/real-time PCR demonstrated that association of RNA polymerase II phospho-S5 C-terminal domain (CTD) repeats was reduced in *Gapdh*, *Pgk1*, and *Ldha* upon deficiency induction (Fig 4C). As they belong to subset "red," these glycolysis enzyme genes are sensitive to a shared function of *Crk* and *Crkl* for their transcription.

## A role for Crk and Crkl in CoCl$_2$-stabilized Hif1a protein pool

Several glycolysis enzymes have been identified as targets of the transcription factor Hif1a (Fig 4A, labels in orange color) (Benita et al, 2009). Although Hif alpha proteins (Hif1a and Hif2a) are rapidly

degraded under the ambient air oxygen level of 21% by the von Hippel–Lindau tumor suppressor VHL and E3 ubiquitin ligase, the degradation process is controlled under physiological O$_2$ levels of 2–9% in tissues and embryonic environment (Simon & Keith, 2008; Semenza, 2017). To investigate possible effects of *Crk/Crkl* deficiency on Hif1 pathways, we used CoCl$_2$ to stabilize Hif1a proteins by inhibiting VHL (Yuan et al, 2003). As anticipated, Hif1a levels increased in the nucleus in the presence of CoCl$_2$ in both *Crk/Crkl* deficiency-induced and uninduced MEFs (Fig 4D). However, the CoCl$_2$-induced increase was much smaller in *Crk/Crkl* deficiency-induced MEFs than that of uninduced control MEFs ($p.adj < 2 \times 10^{-16}$). Although the oxygen-rich environment under the standard tissue culture condition masks Hif1a protein levels, a small difference was also detectable between *Crk/Crkl* deficiency-induced MEFs than that of uninduced control MEFs ($p.adj < 2 \times 10^{-16}$). These results demonstrate that normal Hif1a protein production relies on *Crk* and *Crkl*.

## Crk/Crkl deficiency affects chromatin-level gene regulations

To explore the mechanism by which glycolysis enzyme expression was down-regulated, we conducted genome-wide ChIP-Seq analysis with an active chromatin marker, acetylated histone H3 lysine-27 along with RNA Polymerase II phospho-S5 CTD repeats (H3K27Ac and Pol2, respectively). Association of H3K27Ac and Pol2 with transcription start site (TSS)–proximal regions is a global feature of actively transcribed genes, as H3K27Ac positively enhances the search kinetics of transcription activators as well as the transition of Pol2 from initiation to elongation by accelerating its promoter escape (Stasevich et al, 2014).

H3K27Ac or Pol2 ChIP-Seq showed a positive correlation with mRNA DE for the genes in subset "red" as *Crk/Crkl*-common and *Crk/Crk*-sensitive targets (Fig 5A). In particular, the *down-regulated* glycolysis genes in subset "red" were identified within the lower left quadrant in the scatterplots (Fig 5A). Furthermore, the ChIP-Seq reads for H3K27Ac and Pol2 were reduced globally in *down-regulated* genes, compared with the up-regulated gene group (Fig 5B). Interestingly, the ChIP-Seq reads for H3K27Ac and Pol2 were not increased for subset "*red*" *up-regulated* DE genes with their median values in the negative range. Therefore, reduced mRNA levels of the glycolysis genes were attributable largely to diminished transcription in *Crk/Crkl*-deficiency, whereas a separate mechanism may drive increased steady mRNA levels for the *up-regulated* DE genes in subset "red."

In TSS-flanking regions, we observed generally diminished ChIP-Seq peaks for both H3K27Ac and Pol2 in *Crk/Crkl*-deficient MEFs compared with the control group (Fig 5C, KO versus CTRL in orange and green lines, respectively). Consistent with the result shown in Fig 5B, H3K27Ac and Pol2 signals were not increased for the up-regulated genes in *Crk/Crkl*-deficient MEFs. To quantify the changes, boxplots were generated for the peak height of the ChIP-Seq signals in the TSS ± 2 kb region (Figs 5D and S7). We noted significant differences in H3K27Ac signals between deficiency-induced and uninduced MEFs ($p.adj < 1 \times 10^{-10}$ and $p.adj = 8.97 \times 10^{-04}$ in the *down-regulated* and *up-regulated* gene categories, respectively). Pol2 ChIP-Seq signals were also highly different between deficiency-induced and uninduced MEFs in the *down-regulated* gene

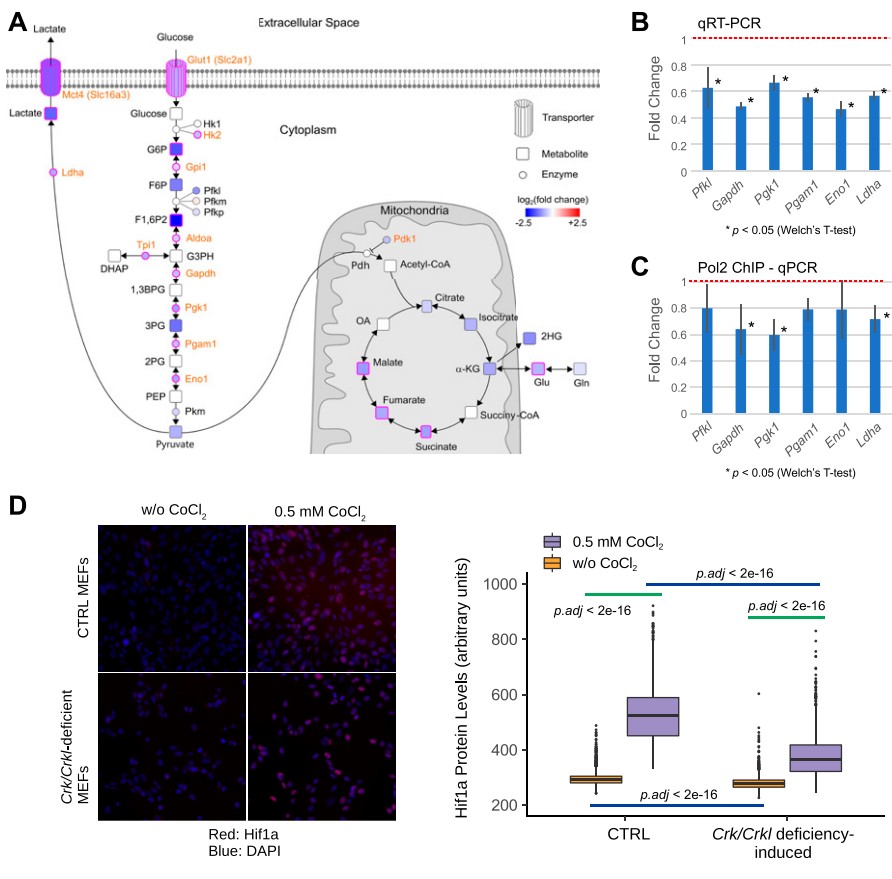

**Figure 4. Glucose metabolism is a common target of *Crk* and *Crkl* deficiency.**
**(A)** CE-TOF/MS metabolome analysis identified differential levels of several metabolites in central carbon metabolism in primary MEFs deficient for either *Crk* or *Crkl*, or for both *Crk* and *Crkl*. The illustration is a compilation of metabolome and RNA-Seq results. Transporters, enzymes, and metabolites affected in *Crk;Crkl* double deficiency are highlighted by a color shade (shades of blue indicate down-regulation; shades of red, up-regulation). Nodes encircled by magenta lines are affected not only in *Crk;Crkl* double deficiency but also in single deficiencies of both *Crk* and *Crkl*. Orange-colored labels indicate known targets of the transcription factor Hif1a. **(B)** Quantitative RT-PCR validated the results of RNA-Seq for several glycolysis genes in primary MEFs deficient for *Crk* and *Crkl*. Levels of expression were expressed as a fold change. The basal level without induction of *Crk;Crkl* deficiency set at 1.0 as shown at the red dotted line. Welch's *t* test was performed on raw Ct values between 4-hydroxytamoxyfen (4OHT)–treated and CTRL groups (n = 3); *P*-values were 0.03462, 0.00061, 0.01735, 0.00027, 0.00834, and 0.00013 for *Pfkl*, *Gapdh*, *Pgk1*, *Pgam1*, *Eno1*, and *Ldha*, respectively. **(C)** Association of RNA polymerase II to several glycolysis genes were decreased in MEFs deficient for *Crk* and *Crkl*. ChIP was conducted with anti-RNA polymerase II hosphor-S5 CTD repeats (Pol2) antibody followed by quantitative PCR. Levels of Pol2 association to each gene were expressed as a fold change. The basal level without induction of *Crk;Crkl* deficiency set at 1.0 as shown at the red dotted line. Welch's *t* test was performed on delta Ct values of chromatin IP samples (relative to their respective DNA input used for IP) between 4-hydroxytamoxyfen (4OHT)–treated and CTRL groups (n = 3); *P*-values were 0.0254, 0.0064, and 0.0177 for *Gapdh*, *Pgk1*, and *Ldha*, respectively. **(D)** Differential Hif1a protein levels were observed in the nucleus between *Crk/Crkl* deficiency–induced and CTRL MEFs with or without CoCl2 to stabilize Hif1 proteins (box plot). Representative images are shown on the left to the box plot. MEFs were incubated with or without 0.5 mM CoCl2 for 4 h before fixation. Hif1a proteins were detected in the IN Cell Analyzer 2000 upon immunofluorescent staining with anti-HIF1A antibody. The nuclei were identified by DAPI staining. Signals in ~2,000–2,300 nuclei were quantified for each group for Hif1a nuclear localization. Kruskal–Wallis tests followed by Dunn's post hoc tests with Bonferroni corrections yielded virtually identical *p*-levels to that of Brunner–Munzel tests.

category (*p.adj* < 1 × 10⁻¹⁰), but not in the *up-regulated* gene category (*p.adj* = 0.964). Pol2 elongation from the TSS downstream beyond +2 kb was also greater for the *down-regulated* genes in control MEFs than that of the *Crk/Crkl* deficiency induced MEFs (Fig 5C, the green versus orange lines in the Pol2 plots). Therefore, down-regulated mRNA levels (found in RNA-Seq) were generally attributable to reduced Pol2 transcription initiation and elongation. On the other hand, up-regulated mRNA expression did not result from elevated promoter activity. These results demonstrate that *Crk/Crkl* deficiency led to widespread H3K27Ac depression in the epigenome, leading to marked reduction in de novo transcription of numerous genes down-regulated as common targets of *Crk* and *Crkl*.

### Crk/Crkl deficiency impairs the effect of glucose on S6K and S6 activation

The results above demonstrated impaired glycolysis in *Crk/Crkl* deficiency accompanied by reduced Hif1a protein production. Signaling pathways known to influence cell size via p70 S6 kinase (S6K encoded by *Rps6kb1* and *Rps6kb2*) and the ribosomal protein S6 (Rps6) are also important for Hif1a translation (Fingar et al, 2002;

Semenza, 2010; Chauvin et al, 2014). We found that glucose availability was essential for maintaining active signaling cascades through Akt, Tsc2, S6K, and S6 in a dose-dependent manner, whereas 5 mM glucose appeared optimal for Akt S473 phosphorylation as well as Tsc2 T1462 phosphorylation (Figs 6A and S8). Upon induction of *Crk/Crkl*-double deficiency, glucose resulted in much muted activation of the cascade compared with that of control MEFs (Fig 6A). Although Akt S473 phosphorylation was reduced in the *Crk/Crkl*-deficient MEFs, glucose-induced Tsc2 T1462 phosphorylation (considered as an Akt-specific phosphorylation site) was greater in the deficiency-induced MEFs than that of the control groups. Because the phosphorylation readout was reduced on both p70 S6K and S6 proteins, these results suggest that intersecting pathways surrounding Akt and Tsc2 may be dysregulated in *Crk/Crkl* deficiency in response to glucose availability.

### Crk/Crkl deficiency and glucose restriction lead to cell membrane blebbing

To provide further evidence for a role that the *Crk* family may play in glucose metabolism, we evaluated the effects of 2-deoxy-D-glucose

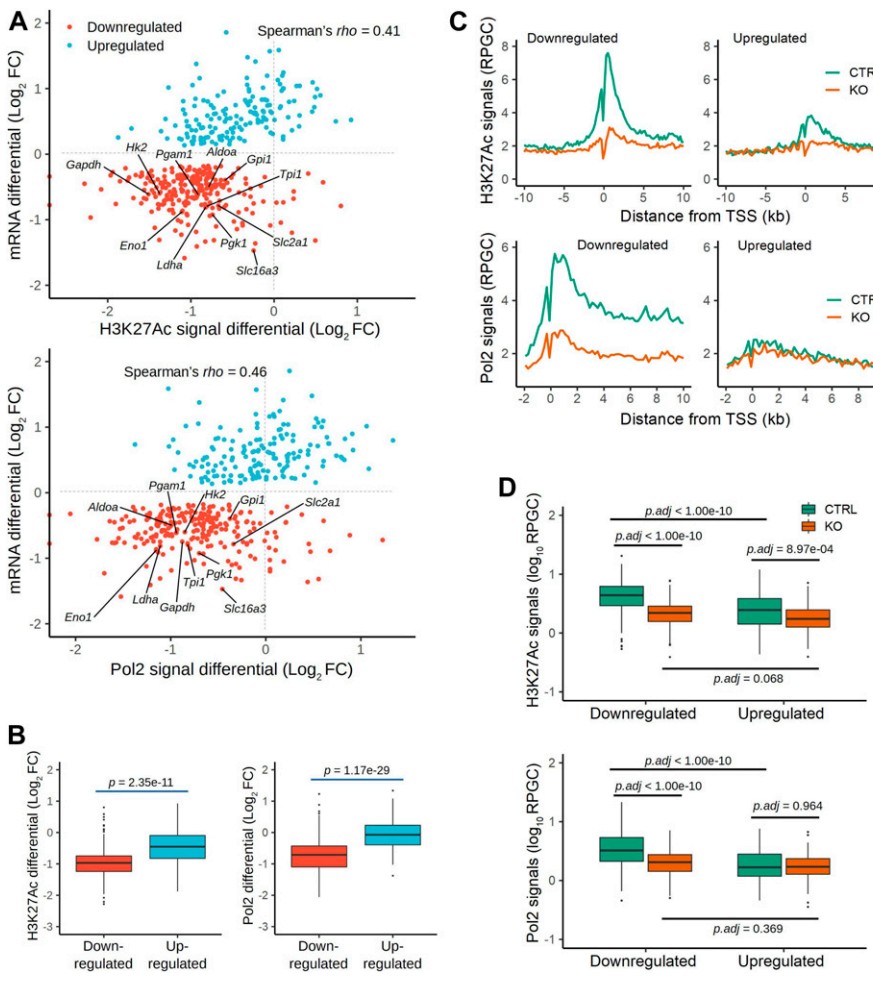

**Figure 5. Chromatin immunoprecipitation (ChIP)-Seq analysis.**
**(A)** Scatterplots indicate ChIP-Seq signals for H3K27Ac and Pol2 (hosphor-S5 CTD repeats) in the x-axis and mRNA levels (data from RNA-Seq) in the y-axis, in which values are shown in log₂ fold change (log₂ FC) as differentials between *Crk/Crkl* deficiency–induced and uninduced MEFs. Each dot represents a single gene identified in subset "red" (Fig 3B). Red or light-blue dots indicate *down-regulated* or *up-regulated* genes based on their expression identified by RNA-Seq, respectively. Glycolytic genes are labeled for their gene symbols. ChIP-Seq signals in these panels are based on peak heights within transcription start site (TSS) ± 2 kb for H3K27Ac and Pol2. Spearman's rank correlation coefficient $\rho$ was calculated for the entire distribution. **(B)** Boxplots indicate distributions of the ChIP-Seq signal differentials within TSS ± 2 kb for H3K27Ac and Pol2 in either *down-regulated* or *up-regulated* categories of subset "red" genes. Whiskers were drawn between the highest and lowest data points within 1.5× interquartile range (IQR) from the upper or lower quartile. Data points outside the 1.5× IQR are indicated as outliers (dots). The *P*-values were calculated by Mann–Whitney U tests between *down-regulated* and *up-regulated* categories. **(C)** XY plots indicate the average ChIP-Seq signals as reads per genome content (RPGC) in the y-axis and the distance from TSS in the x-axis in *Crk/Crkl* deficiency-induced and uninduced MEFs (KO and CTRL) in orange and green lines, respectively. Note that when ChIP-Seq signals are compared in CTRL MEFs between the down-regulated and up-regulated gene groups, the down-regulated group shows greater average peak heights in both H3K27Ac and Pol2 ChIP-Seq signals. **(D)** Boxplots indicate the distribution of the ChIP-Seq signals (RPGC) within TSS ± 2 kb in *Crk/Crkl* deficiency-induced and uninduced MEFs (KO and CTRL), for genes *down-regulated* or *up-regulated* in subset "red." The y-axis is in a log₁₀ scale of RPGC. ANOVA and Tukey post hoc tests were performed on log₁₀-transformed RPGC values to bring the data distributions closer to Gaussian distributions. See Fig S7 for boxplots using untransformed data and square-root transformed data.

(2DG), a competitive inhibitor of glucose metabolism, on *Crk/Crkl* deficient or control MEFs (Gu et al, 2017). We noted that when cultured in a glucose-controlled condition (5 mM glucose with 10% dialyzed FBS), a range of 2DG concentrations induced blebbing cell morphology identified by cell staining with CellMask, DAPI, and anti-vinculin (Fig S9). "Blebbing" is a feature characterized by several plasma membrane protrusions resembling small beads that decorate the cell edge boundary, as a stage of apoptotic or non-apoptotic processes (Coleman et al, 2001; Fackler & Grosse, 2008). To minimize the subjective nature of categorical judgment on cell morphology, we applied an automated computational analysis. Using a few parameters standardized for blebbing identification (Fig S9), we found that *Crk/Crkl* deficiency exacerbated the blebbing phenotype induced by 2DG (Fig 6B), thus consistent with their possible involvement in glucose metabolism. Although the results do not distinguish apoptotic versus nonapoptotic blebbing, it is more likely that 2DG-induced blebbing may be an indication of apoptosis based on the observation that 2DG treatments reduced the total cell counts in our experimental condition (see the "n" numbers on top of each bar in Fig 6B).

## A role for Crk and Crkl in IGF1-induced S6K/S6 activation

Insulin-like growth factor 1 (Igf1) is one of the growth factors required for normal development and known to control cell size through Akt (Lloyd, 2013; Manning & Toker, 2017). Igf1 signaling was implicated in our global analysis of the RNA-Seq results (Fig 3A), and *Igf1* was one of the up-regulated DE genes in subset "red" in Fig 3B (see also Supplemental Data 1). Real-time quantitative RT-PCR using two non-overlapping sets of *Igf1*-specific primers confirmed that steady-state *Igf1* mRNA levels were increased nearly 10-fold upon induction of *Crk/Crkl* double deficiency compared with the MEFs without deficiency induction (Fig 6C). Despite this up-regulated *Igf1* expression, *Crk/Crkl* double deficiency–induced MEFs exhibited muted responses to IGF1 for activating S6K and S6 (Fig 6D). These results demonstrate that *Crk/Crkl* deficiency uncouples the autocrine/paracrine growth factor Igf1 from transducing S6K-S6 activation. Interestingly, however, *Crk/Crkl* deficiency did not inhibit Igf1-induced Akt S473 phosphorylation, suggesting that Igf1 was able to activate MTORC2 complex. In addition, overexpression of *Crk* or *Crkl* by itself increased both phosphorylation and protein levels of S6 in

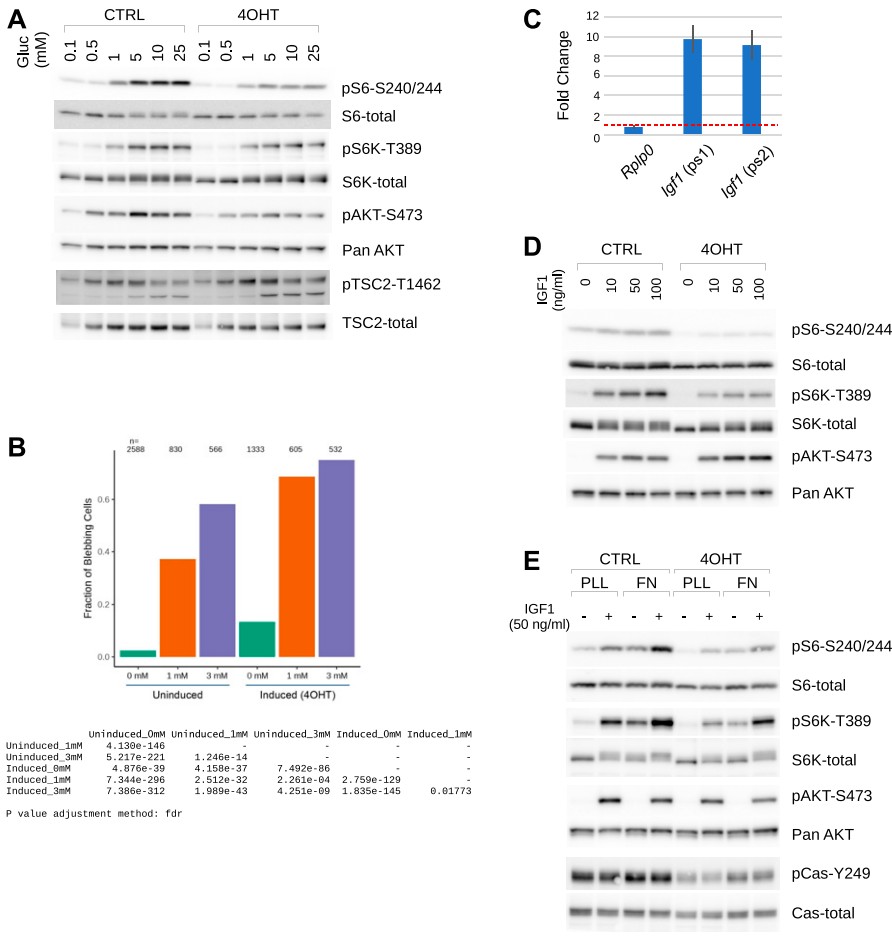

**Figure 6. Deficiency for *Crk* and *Crkl* results in aberrant glucose metabolism and *Igf1* signaling.** **(A)** Glucose availability in culture media throttled dose-dependent phosphorylation of ribosomal protein S6 (pS6-S240/244), as well as that of Akt (pAKT-S473), TSC2 (pTSC2-T1462), and p70 S6 kinase (pS6K-T389) as shown in immunoblots. MEFs were cultured with indicated concentrations of glucose for 24 h after a 24-h period of glucose restriction at a concentration of 0.1 mM, the lowest glucose concentration that MEFs can tolerate in the presence of 10% dialyzed FBS (Fig S8). As a point of reference, basal DMEM includes 5 mM glucose, whereas a high-glucose formula includes 25 mM glucose. **(B)** The glucose metabolism inhibitor 2-deoxy-D-glucose (2DG) induced a cell blebbing phenotype in a dose-dependent manner, and *Crk/Crkl* deficiency significantly exacerbated the frequency of the phenotype in single cell analysis. The 2DG concentrations are indicated under the x-axis. The frequency of the blebbing phenotype was determined as described in the Materials and Methods section and Fig S9. Large numbers of cells (n) were imaged in a high-content imaging apparatus, and individual cells were analyzed through a series of MATLAB scripts. Beneath the bar graph is a matrix table for *P*-values by Fisher's exact tests adjusted for multiple comparisons (FDR). **(C)** *Crk/Crkl* deficiency increased *Igf1* mRNA levels. The bar graph represents levels of *Igf1* messages upon *Crk/Crkl* deficiency induction in MEFs, relative to that of their controls without deficiency induction. As a control, fold change of *Rplp0* (60S ribosomal protein p0) is also shown. Two non-overlapping primer sets 1 and 2 (ps1 and ps2) were used to confirm up-regulation approximately at an order of magnitude greater. Error bars indicate standard deviations (n = 3). Red-dotted line shows the level in control samples set at a relative fold change of 1. **(D)** *Crk/Crkl* deficiency resulted in muted response to exogenous *Igf1* over a range of doses in phosphorylation of p70 S6 kinase (S6K) and ribosomal protein S6. In contrast, Akt phosphorylation was induced by Igf1, suggesting a role of Crk and Crkl in a regulatory mechanism on S6K and S6. MEFs were cultured without serum for the last 3 h of 48 h post deficiency induction, then were stimulated with Igf1 at different concentrations indicated for 15 min. **(E)** Crk and Crkl played an important role in S6K and S6 phosphorylation on which both Igf1 and fibronectin (FN) cooperate as shown in immunoblots probed with different antibodies. Note that FN alone did not activate Akt (as seen in pAkt-S473), whereas Igf1 did. After 3 h serum starvation, MEFs were re-plated on FN or poly-L-lysine in serum-free medium for 60 min followed by incubation with Igf1 for 15 min before harvest.

HEK293 cells, whereas little effects were observed on p70[S6K] (Fig S10). Because p70[S6K] lies upstream of S6, these results suggest the possibility that Crk and Crkl may play a role in Igf1-induced S6K and S6 activation in addition to the canonical Akt-mediated pathway, while also implicating Crk and Crkl in a mechanism by which they may activate S6 independent of p70[S6K].

## Cooperative signaling between Igf1 and integrins

Crk and Crkl are involved in a broad range of signaling pathways associated with tyrosine kinases (Feller, 2001; Birge et al, 2009). Among them are pathways mediated by extracellular matrix (ECM) proteins and integrins. We, therefore, investigated pathway integration between Igf1 and the ECM protein fibronectin (FN) for activating p70[S6K] and S6 in MEFs (Fig 6E). Among the deficiency-uninduced control groups (Fig 6E left half, the CTRL lanes), Igf1 induced phospho-Akt S473 in 15 min at comparable levels between the poly-L-lysine or FN groups (PLL or FN, respectively), whereas neither PLL nor FN induced Akt phosphorylation without Igf1. We noted that

without Igf1, phosphorylation of S6 (S240/244) and S6K (T389) was increased by plating on FN compared with that of PLL, thus the ability of FN to increase phosphorylation of S6 and S6K appears to be independent of Akt. When Igf1 stimulated the MEFs plated on FN, the phosphorylation levels on S6 and S6K was highest among the uninduced groups. Among the deficiency-induced groups (Fig 6E right half, the 4OHT lanes), we observed that although the general trend is similar to the deficiency-uninduced control groups (the CTRL lanes), the levels of S6 and S6K phosphorylation decreased in the 4OHT groups for their responses to Igf1 and FN, independently or combined. These results demonstrate an important role of Crk and Crkl in mediating cooperative signals to S6K and S6 activation from Igf1 and FN.

## Rescue of Crk/Crkl deficiency by an activated Rapgef1

*Rapgef1* (also known as *C3G*) encodes a guanine–nucleotide exchange factor for the small G-protein Rap1 (encoded by *Rap1a* and *Rap1b*) as one of the major proteins to which the SH3n domain of

Crk and Crkl can associate (Feller, 2001; Birge et al, 2009). *Rapgef1* is ubiquitously expressed during early-mid gestation mouse embryos and its genetic ablation results in an early embryonic phenotype at E7.5, whereas a hypomorphic mutation generates a vascular phenotype around E11.5-E14.5 (Ohba et al, 2001; Voss et al, 2003). These reports also demonstrated that Rapgef1 is an important mediator of cell adhesion to ECM proteins associated with reduced numbers of focal adhesions in MEFs isolated from the mutant embryos. We found that an activated Rapgef1 (C3GF) conferred MEFs resistance to *Crk/Crkl* deficiency for cell size (Fig 7A). C3GF also rescued *Crk/Crkl*-deficient MEFs for cell proliferation (Fig 7B) and restored expression of some glycolysis enzyme genes that were down-regulated in *Crk/Crkl* deficiency (Fig 7C). *Crk/Crkl* deficiency–induced reduction of fructose-1,6-bisphosphate (F1,6P2) was also restored by C3GF (Fig 7D). C3GF increased S6 and S6K phosphorylation, accompanied by elevated Akt phosphorylation (Fig 7E). Interestingly, C3GF by itself elevated tyrosine phosphorylation and protein levels of p130$^{Cas}$/Bcar1, events thought to be upstream of Rapgef1, compared with that of the vector control groups. Likewise, C3GF enhanced focal adhesions as identified by subcellular localization of total phosphotyrosine, phosphorylated p130$^{Cas}$/Bcar1, and the Fak. On the other hand, *Crk/Crkl*-deficient MEFs in the vector control group appeared to have fewer focal adhesions (Fig 7F). As cells exhibited varying numbers of focal adhesions in each group, evaluating a small number of cells may introduce unintended bias. To objectively quantify focal adhesions in a large number of cells, we adopted an automated image analysis (Fig S11). As anticipated, whereas the number of focal adhesions was reduced by *Crk/Crkl* deficiency, C3GF expression normalized focal adhesion counts (Fig 7G). These results confirmed not only the role of Rapgef1 in mediating positive-feedback signals from Crk and Crkl but also its important functions in glucose metabolism and cell size/adhesion homeostasis.

# Discussion

Our present study has demonstrated that compound heterozygosity of *Crk* and *Crkl* (loss of shared functions) as well as individual gene disruption can generate developmental defects in mice, part of which resemble DiGeorge anomaly in multiple aspects, despite the fact that *CRK* is not a 22q11 gene. Furthermore, *Tbx1* genetic interaction with not only *Crkl* but also with *Crk* provides evidence for a possible functional intersection among these genes. We have demonstrated that normal mesoderm requires at least 50% of the *Crk* family-combined dosage (Fig 1). It is noteworthy that *Tbx1* is essential in the mesoderm for normal heart and outflow tract development, whereas *Tbx1* expression is also required in the epithelia of ectoderm or endoderm origins for normal fourth arch artery and thymic development (Zhang et al, 2006). *Tbx1* knockdown in a cardiomyocyte-differentiating P19 subline as well as *Tbx1*-mutant embryos show abnormal histone H3 monomethyl-K4 profiles (Fulcoli et al, 2016). It is also noteworthy that *Tbx1* deficiency causes DE in mTOR signaling pathway, VEGF signaling pathway, phosphatidylinositol signaling pathway and focal adhesion (Fulcoli

et al, 2016), which we have also identified as *Crk/Crkl*-shared pathways in this study (Fig 3A and C). In fact, *Tbx1* knockdown results in a reduced number/size of focal adhesions in C2C12 cells (Alfano et al, 2019), in similar ways to *Crk/Crkl*-deficient MEFs, we analyzed in this study (Fig 7). Taken together, *Crk*, *Crkl*, and *Tbx1* may regulate the gene regulatory network by modulating global epigenetic landscape, which directly or indirectly control cell behavior through cell–matrix adhesion and metabolism.

Our results have implicated *Crk* and *Crkl* in glucose metabolism through the transcription factor Hif1a. Whereas hypoxic conditions are known to increase Hif1 protein levels by stabilization, Hif1a is essential for developmental processes under physiological oxygen levels of 2–9% $O_2$ in mouse embryos (Carmeliet et al, 1998; Iyer et al, 1998; Ryan et al, 1998). Furthermore, *Hif1a* is required for normal expression of several glycolytic enzyme genes such as *Glut1*, *Pfkl*, *Aldoa*, *Tpi1*, *Gapdh*, *Pgk1*, and *Ldha* under the ambient oxygen level as well as in 1% $O_2$ in mouse embryonic stem (ES) cells (Iyer et al, 1998; Ryan et al, 1998). Therefore, impaired Hif1a protein production may be attributable to reduced glycolysis gene expression in *Crk/Crkl*-deficient MEFs, although investigated in the ambient oxygen level (Fig 4). Many MTORC1-inducible genes have been identified with Hif1- and Myc-binding sites, whereas Hif1a is essential for MTORC1-dependent glycolytic gene expression (Düvel et al, 2010). It was also reported that Myc stabilizes HIF1a post-translationally and that *Myc*-induced transformation requires Hif1a in the human immortalized mammary cell line IMEC in normoxia (Doe et al, 2012). We noted that *Myc* was one of the down-regulated genes in subset "red," and our ChIP-Seq results also indicated reduced association of H3K27Ac and Pol2 markers with *Myc* in *Crk/Crkl*-deficiency induced MEFs (Supplemental Data 2 and Fig S12).

Vascular endothelial growth factor A (*VEGFA*) is one of the targets of Hif1 (Forsythe et al, 1996). It has been reported that IGF1 can stimulate *VEGFA* mRNA expression by stabilizing HIF1A protein in human colon cancer cell line HCT116 (Fukuda et al, 2002). We noted that *Vegfa* was down-regulated in subset "red," thus commonly affected by *Crk* and *Crkl* (Supplemental Data 2). Analysis of the ChIP-Seq signals around the *Vegfa* gene revealed that its promoter-proximal region was poorly associated with H3K27Ac and Pol2 CTD phospho-S5, thus indicating that *Vegfa* promoter activity was suppressed in *Crk/Crkl*-deficient MEFs (Fig S12). Although *Igf1* deficiency has not been linked to DiGeorge-like anomaly in humans or in animal models, a positive role for Igf1 has been demonstrated in promoting mesoderm development and vasculogenesis in mouse embryoid bodies (Piecewicz et al, 2012). *Vegfa* is known as a dosage-sensitive gene for normal development and *Vegfa*$^{164}$-isoform deficiency results in DiGeorge-like anomaly in mice (Carmeliet et al, 1996; Ferrara et al, 1996; Stalmans et al, 2003). Reduced *vegfa* also shows genetic interactions with *Tbx1* knockdown in zebrafish (Stalmans et al, 2003). Therefore, reduction of *Vegfa* expression may also contribute to an impaired genetic network in which *Crk* and *Crkl* may have common intersection with *Tbx1*.

In this study, we have focused on the genetic and epigenetic network down-regulated in *Crk/Crkl* deficiency because up-regulated genes found in RNA-Seq may not be effectively translated into increased protein productions because of suppressed p70$^{S6K}$/S6 activities in *Crk/Crkl*-deficient MEFs. Although this study did not

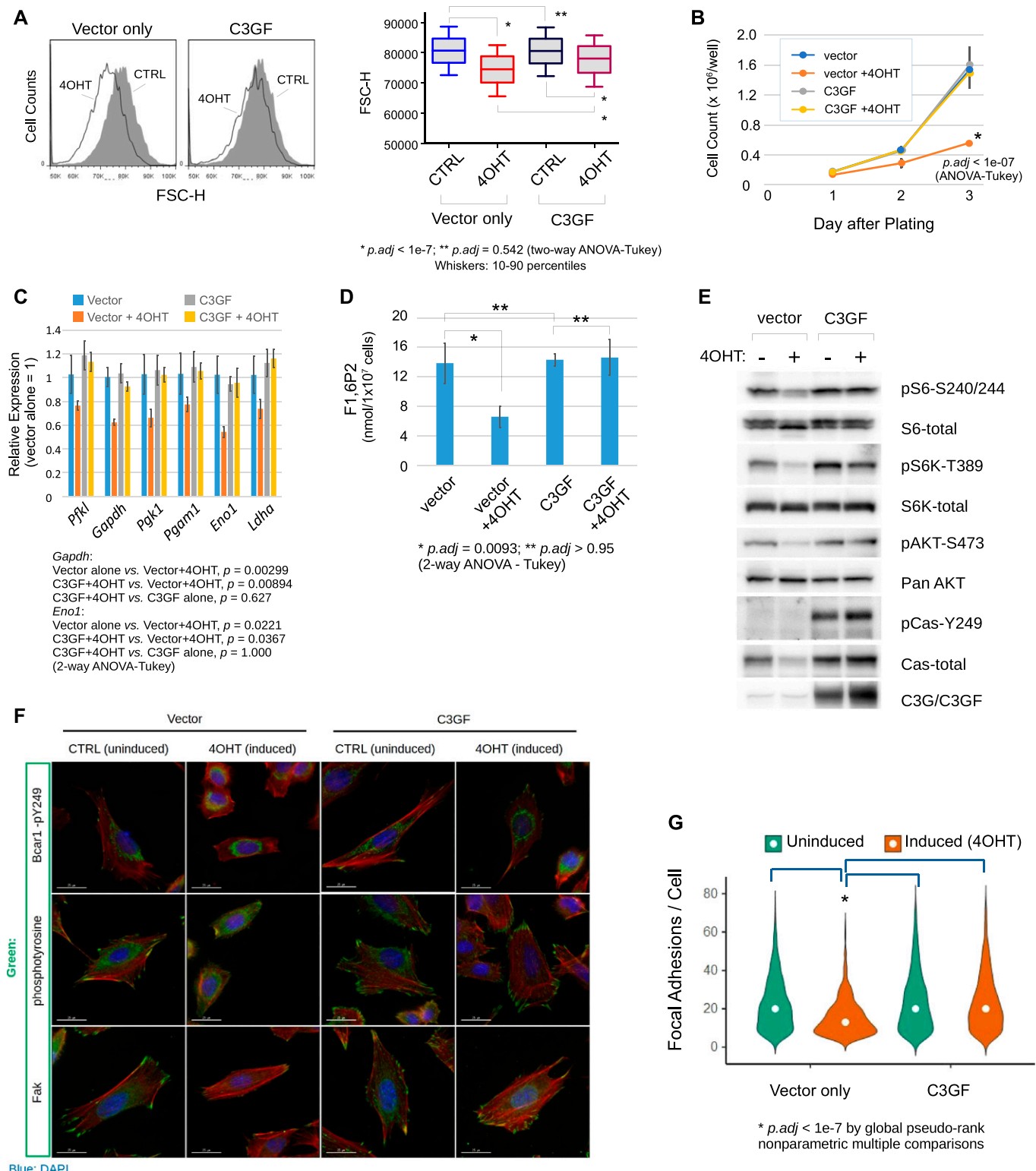

**Figure 7. An activated Rapgef1 partially rescues aspects of *Crk/Crkl* deficiency in MEFs.**
**(A)** Overexpression of C3GF partially blocked *Crk/Crkl*-deficiency–induced cell size changes. The histograms show distributions of cell sizes as estimated by FSC-H measurements in FACS analysis of ~6,000–7,000 cells in the G1 phase in each group. To compare the distributions, a boxplot was generated below the histograms. Human *RAPGEF1* fused to a farnesylation sequence (C3GF) or empty vector was introduced into MEFs before *Crk/Crkl* deficiency induction by 4OHT. Two-way ANOVA followed by Tukey post hoc tests were performed for statistical comparisons. **(B)** Overexpression of C3GF rescued Crk/Crkl deficiency–induced inhibition of cell proliferation. Cell

investigate precise mechanisms that underlie the abnormal cell contact behavior (Fig 2B–D), a recent study reported that normal CIL relies on Fak and Src for coordinated redistribution of cell–matrix contacts and intracellular force in frog neural crest cells (Roycroft et al, 2018). Because Crk and Crkl mediate signals partly from Fak and Src (Shin et al, 2004; Birge et al, 2009; Watanabe et al, 2009), it is plausible that Crk and/or Crkl are also required for traction force redistribution, a key mechanism for repulsive locomotion as an essential feature of mesenchymal cells. In this regard, it is noteworthy that *Crk/Crkl* deficiency inhibits focal adhesions functionally and structurally (Figs 3A and C and 7F and G). Focal adhesions include several mechanosensor proteins such as the Crk/Crkl SH2-binding protein p130$^{Cas}$ (Bcar1), and the small G-protein Rap1 has been identified as a critical participant in mechanotransduction and mechanosensing (Sawada et al, 2006; Lakshmikanthan et al, 2015; Freeman et al, 2017). Because an activated mutant of the Crk/Crkl SH3-binding protein Rapgef1 (C3G) rescues *Crk/Crkl* deficiency for glycolysis, cell proliferation, and focal adhesions in MEFs (Fig 7), the Crk/Crkl-Rapgef1-Rap1 axis is a modular pathway which senses multiple extracellular signals such as Igf1 and integrin-dependent mechanotransduction. While *Crk/Crkl* deficiency exacerbated the blebbing phenotype induced by 2DG, the phenotype was different from the focal adhesion phenotype in *Crk/Crkl*-deficient MEFs (Figs 6B and 7F and G). Because activation of Rapgef1-rescued cell size/glucose metabolism as well as focal adhesions in *Crk/Crkl* deficiency, it is tempting to hypothesize that glucose metabolism may be regulated downstream of focal adhesions/cell–matrix adhesion.

It is also noteworthy that delayed postnatal growth is common among DGS/22q11.2DS patients (McDonald-McGinn et al, 2015). A recent study has reported that a 22q11.2DS patient with a small statue had growth-hormone and IGF1 deficiency (Bossi et al, 2016). Therefore, impaired responses to IGF1 that we found in MEFs may also have clinical relevance. In addition, although the number of reported cases are relatively few, maternal diabetes has been linked to thymic and kidney defects associated with tetralogy of Fallot and other congenital disorders in infants without a deletion in 22q11.21 (Novak & Robinson, 1994; Digilio et al, 1995; Cirillo et al, 2017; Taliana et al, 2017). We speculate that maternal glucose metabolism may be a possible contributing factor that could partly explain large variations of penetrance and expressivity observed among 22q11.2DS patients. Future studies are warranted to investigate the mechanisms by which the cell-adhesion signaling axis involving Crk/Crkl, and Rapgef1 regulates the epigenetic network important for

metabolism and proper development of tissues affected in DiGeorge/22q11.2DS patients.

# Materials and Methods

### Generation of *Crk* conditional knockout mice

The mouse *Crk* gene was targeted in an 129S6-derived ES cell line using a homologous recombination vector assembled using genomic fragments isolated from an 129-derived genomic library as well as *FRT-PGKneo-FRT* (*FneoF*) and *loxP* sequences as illustrated in Fig S1. Targeted ES cells were injected into C57BL/6J blastocysts to generate chimeric mice via standard technique in the Transgenic and ES Cell Technology Mouse Core Facility at the University of Chicago. Highly chimeric animals were then backcrossed with C57BL/6J. The *PGKneo* cassette was removed by a cross with the *FLPeR* mice (B6.129S4-Gt(ROSA)26Sor$^{tm1(FLP1)Dym}$). Mice heterozygous for *Crk* and *FLPeR* was then backcrossed with C57BL/6 to segregate out *FLPeR*. *Crk* heterozygous mice without *neo* or *flp* were then selected as a knockout-ready strain, *Crk* $^f$ (*Crk-floxed exon 1*; B6.129S4-Crk$^{tm1.1Imo}$/J). We previously generated and reported a *Crkl* conditional strain (B6;129S4-Crkl$^{tm1c(EUCOMM)Hmgu}$/ImoJ) (Haller et al, 2017; Lopez-Rivera et al, 2017). To make distinction easier from *Crk* $^f$, we call the *Crkl* knockout-ready strain *Crkl* $^{f2}$ because *Crkl* exon 2 is flanked by two *loxP* sites. After more than five generations of backcross with B6, some *Crk* $^f$/+ and *Crkl* $^{f2}$/+ mice were crossed with *R26 Cre-ER$^{T2}$* strain (B6.129-Gt(ROSA)26Sor$^{tm1(cre/ERT2)Tyj}$/J) or with *Mesp1$^{Cre}$* (Saga et al, 1999) to set up 4-hydroxytamoxifen (4OHT)-inducible or mesoderm-specific knockouts, respectively. For some experiments, *Crk$^d$* or *Crkl$^{d2}$* (deletion of *Crk* exon 1 or *Crkl* exon 2, respectively) was generated as a knockout allele by crossing the knockout-ready strains and a global-deletion strain, *Meox2$^{Cre}$* (B6.129S4-Meox2$^{tm1(cre)Sor}$/J) (Tallquist & Soriano, 2000). *Meox2$^{Cre}$* was then segregated out by backcross with C57BL/6J, and *Crk$^d$* and *Crkl$^{d2}$* heterozygous mice were maintained by continual backcross with C57BL/6J. In some experiments, *Crk$^d$* were crossed with *Tbx1$^-$* heterozygotes (a gift from Virginia Papaioannou) had been maintained by continual backcross with C57BL/6J more than 11 generations. Mouse embryos were isolated at various stages of development by timed mating. Mice and embryos were genotyped using PCR primers listed in Table S3. All mouse works were carried out in strict accordance with the protocols approved by the Institutional Animal Care and Use Committee of the University of Chicago.

---

numbers were counted in tissue culture plates for 3 d after plating. Bars indicate standard deviations from triplicate determinations (n = 3). Two-way ANOVA followed by Tukey post hoc tests were performed for statistical comparisons. **(C)** C3GF restored expression of the glycolytic enzyme genes. Expression of glycolytic genes were determined in real-time/quantitative RT-PCR. Bars indicate standard deviations. Two-way ANOVA followed by Tukey post hoc tests were performed on raw Ct values (n = 3). **(D)** C3GF blocked Crk/Crkl deficiency from reducing the level of fructose-1,6-bisphosphate (F1,6P2). Bars indicate standard deviations from triplicate determinations (n = 3). Two-way ANOVA followed by Tukey post hoc tests were performed for statistical comparisons. **(E)** Immunoblots show C3GF-dependent rescues on S6, S6K, and Akt phosphorylation associated with elevated phosphorylation of the focal adhesion protein p130$^{Cas}$ (Bcar1). **(F)** C3GF restored phosphorylated Bcar1, phosphotyrosine, and Fak localization at focal adhesions. Representative fluorescent microscopy images are shown. **(G)** Violin plots show quantitative results of focal adhesions. Focal adhesions were identified by localization of Fak in immunostained MEFs followed by automated image acquisition and analysis as described in the Materials and Methods section (also see Fig S11). The sample size (the number of cells per group) was 1,481, 516, 1,666, and 1,374 in a 2X2 experimental design (Uninduced/Vector Only, 4OHT-Induced/Vector Only, Uninduced/C3GF, and 4OHT-Induced/C3GF groups, respectively) after applying the cutoffs indicated in Fig S11 to minimize the possibility of counting artifacts in staining and segmentation. However, inclusion of all cells without cutoffs did not affect the statistical outcome. Statistical analysis was performed by a global pseudo rank method with Tukey tests adjusted for multiple comparisons using the *mctp* function in the *nparcomp* package written in the programming language *R* (Konietschke et al, 2015). Similar statistical outcome was obtained by two-way ANOVA after log$_{10}$-transformation (Fig S11D). White circles in each violin plot indicate the position of the median.

## RNA in situ hybridization

Anti-sense RNA probes were generated from pBluescript plasmids that included ~700-bp fragments isolated from the 3′ UTR of mouse *Crk* and *Crkl* cDNAs. The full-length cDNAs were synthesized by RT-PCR using oligo(dT) and gene-specific 5′ UTR primers using total RNA isolated from C57BL/6J E10.5 embryos. The cDNA sequences were confirmed by low-throughput Sanger sequencing from plasmid primers. RNA in situ hybridization was carried out in E10.5 mouse embryos isolated from C57BL/6J mice as previously described (Guris et al, 2006).

## MEFs

Primary MEFs were isolated from individual embryos at E11.5 and cultured in DMEM high-glucose formula supplemented with 0.1 mM 2-mercaptoethanol and 10% fetal bovine serum (HyClone) as previously described (Li et al, 2002). Embryos and MEFs were dissociated using Accutase or TrypLE (Thermo Fisher Scientific). MEFs were split 1:3 for maintenance every 3 d. Cre-mediated gene deficiency was induced by 0.25 µM (Z)-4-hydroxytamoxifen (Sigma-Aldrich) for 24 h in MEFs having a genetic background of *R26 Cre-ER^{T2}*, and then washed and replated 1:3 into new plates. Cells were harvested 48 h after removal of 4OHT as deficiency-induced MEFs for experiments, unless otherwise indicated. In some experiments, glass coverslips or culture plates were coated with 0.1% gelatin (porcine skin, Sigma-Aldrich), fibronectin (bovine plasma; Sigma-Aldrich), or poly-L-lysine (Sigma-Aldrich) before experimental replating.

For some experiments, MEFs were stimulated with recombinant human IGF1 (291-G1; R&D Systems) for 15 min after a short serum starvation period of 3 h (longer serum-free starvation caused apoptosis in *Crk/Crkl* double-deficient cells). To determine the effect of medium glucose concentrations, MEFs were incubated with glucose-free DMEM supplemented with various concentrations of glucose and 10% dialyzed FBS after glucose deprivation down to 0.1% for 24 h. The starvation concentration of glucose was determined as shown in Fig S8.

For measurements of cell spreading, MEFs were dissociated with Accutase and suspended in serum-free DMEM, then plated on gelatin-coated plates at a low density so that most cells do not contact each other. Cells were fixed at each time point, and only adherent cells were pictured under a 10× objective after wash. The number of pixels that each cell occupied were determined in eight most spread cells selected per field in three randomly selected fields per plate, in three plates per time point per group using ImageJ (thus, each data point represents a collection of data from a total of 72 cells).

To estimate cell size, light scatters (FSC-H, FSC-A, SSC-H, and SSC-A) were measured in a fluorescence-activated cell sorting machine (FACS Canto II; BD Bioscience), after fixing cells with ethanol and stained using PI/RNase staining buffer (550825; BD Pharmingen). ~6,000 or more cells were measured in each group.

## Transfection and viral transduction

To transfect or infect MEFs for transducing exogenous transgene expression, primary MEFs were kept on the 3T3 protocol until their proliferation was easily maintained and, therefore, considered spontaneously immortalized (passage 15 or greater). To generate MEFs that express an activated RAPGEF1 (C3G), a full coding sequence of human C3G fused to the RAS farnesylation site (C3GF, a gift from Michiyuki Matsuda) was subcloned into pMX-ires-GFP vector for retrovirus production (pMX-C3GF-ires-GFP). Ecotropic retrovirus was generated in Plat-E packaging cells and used to infect immortalized MEFs per standard protocols. Control MEFs were generated with pMX-ires-GFP without C3GF. GFP-positive cells were then selected by FACS and maintained for experiments. In some experiments, in-frame fusions of EGFP and human CRK or CRKL was constructed using pEGFPC2 plasmid (Clontech-TAKARA). Human embryonic kidney 293 cells were transfected with the plasmid to overexpress EGFP-CRK or CRKL using Lipofectamine LTX (Invitrogen) as recommended in the manufacturer's protocol.

## Immunofluorescence staining

For detection of Hif1a proteins, MEFs induced for *Crk/Crkl* deficiency were replated in 96-well plates at a density of $4 \times 10^4$ cells/well 24 h before harvest (the time of harvest was 72 h from the time 4OHT was added as described above). Some cells were treated with CoCl$_2$ at a final concentration of 0.5 mM for 4 h before fixation with 2% paraformaldehyde. Cells were permeabilized with 0.1% Triton X-100 for 5 min and blocked with 10% FBS and Blocking One (Nacalai). Hif1a was detected with mouse monoclonal anti-Hif1a antibody clone H1alpha67 (NB100-123; Novas) and goat anti-mouse IgG conjugated with Dylight 549 (Thermo Fisher Scientific). Nuclei were counter-stained with DAPI. Fluorescent signals were detected in IN Cell Analyzer 2000 (GE Healthcare).

For staining other cellular proteins, MEFs were replated on glass coverslips coated with 0.1% gelatin. 24 h after replating, MEFs were fixed for 15 min with 4% paraformaldehyde, 1.5% BSA fraction V, and 0.5% Triton X-100 in 1× CB cytoskeletal buffer (10 mM MES, pH 6.8, 3 mM MgCl$_2$, 138 mM KCl, and 2 mM EGTA). After three washes, the cells were incubated with the primary antibody (mouse monoclonal anti-FAK, clone 4.47, 05-537; EMD-Millipore; rabbit anti-p130CAS phospho-Y249, #4014; Cell Signaling Technology; or mouse monoclonal anti-phosphotyrosine, 4G10, 05-321; EMD-Millipore) diluted 1:200 in 1× CB buffer containing 1.5% BSA and 0.5% Triton X-100 for 1 h. After three washes, the cells were incubated with a Dylight 550–conjugated secondary antibody (Thermo Fisher Scientific) that matches the species specificity of the primary antibody diluted 1:1,000 in 1× CB buffer containing 1.5% BSA and 0.5% Triton X-100 for 1 h. F-actin and nuclei/DNA was stained with Alexa Fluor 647 phalloidin and DAPI (Thermo Fisher Scientific) according to a standard staining method. Stained cells were mounted in Prolong Gold Antifade (Thermo Fisher Scientific) and observed under a 60× oil objective lens in DeltaVision Elite deconvolution microscope system (GE Healthcare). In some experiments, MEFs were replated with or without induction of *Crk/Crkl* deficiency in a gelatin-coated 96-well plate and stained as above for high-throughput image acquisitions in IN Cell Analyzer 2500HS (GE Healthcare). For such experiments, additional counterstaining was performed using HCS CellMask Deep Red (Invitrogen/Thermo Fisher Scientific) for identification and segmentation of the cell body (see also the Automated Image Analysis section below).

## Immunoblot

Cell lysates were prepared with lysis buffer containing 1% NP-40 (or IGEPAL CA-630), 50 mM Tris pH 7.5, 10% glycerol, 0.2 M NaCl, 2 mM MgCl$_2$, cOmplete protease inhibitor cocktail (Roche) and PhosSTOP (Roche). Immunoblots were prepared on Immobilon-P membrane (EMD-Millipore) after electrophoresis in SDS-polyacrylamide gel (7.5–15% gradient) using a standard protocol. Proteins were then detected using the following primary antibodies: anti-phospho-S6 S240/244 (#2215; Cell Signaling Technology), anti-S6 (CST#2217), anti-phospho-p70 S6K T389 (CST#9205), anti-p70 S6K (CST#9202), anti-phospho-AKT S473 (CST#4060), anti-pan AKT (CST#2920), anti-phospho-TSC2 T1462 (CST#3617), anti-TSC2 (CST#3990), anti-phospho-p130$^{CAS}$ Y247 (CST#4014), anti-p130$^{CAS}$ (BD 610272), anti-CRK (BD 610035), anti-CRKL (05-414; EMD Milllipore), and anti-C3G (sc-15359; Santa Cruz Biotechnology). Using a horseradish peroxidase–conjugated secondary antibody matching the species of the primary antibody, chemiluminescence was detected on the immunoblot in Image-Quant LAS4000 (GE Healthcare).

## RNA-Seq

RNA-Seq analysis was conducted using total RNA isolated from primary MEFs. Four embryos were isolated for each genotype (*Crk* $^{f/f}$, *Crkl* $^{f2/f2}$, *Crk* $^{f/f}$;*Crkl* $^{f2/f2}$, or wild-type; all compound heterozygous for R26creERT2) as four independent samples per genotype, with an exception that we isolated only two wild-type embryos as negative control samples. When cells were subconfluent, each cell lot was then induced or uninduced for deficiency with 4-hydroxytamoxifen for 24 h, then replated on to new plates without 4OHT to expand for 48 h before harvest. Total RNA was isolated using a Qiaquick RNA isolation kit as described in the manufacturer's protocol. The quality of isolated RNA was checked in a 2100 Bioanalyzer (Agilent Technologies). RNA sequencing was performed in an Illumina HiSeq 2000 with paired end reads. The average inner fragment size was ~250 bp. The sequence reads were filtered by PRINSEQ version 0.20.4 for sequence data quality control (Schmieder & Edwards, 2011), then mapped to the mouse genome sequence in GRCm38.p3 using Tophat2 (version 2.1.0) with the following parameters: –mate-inner-dist 250 –mate-std-dev 40 (Kim et al, 2013). Aligned read counts assigned to RefSeq annotations were obtained by the featureCounts (version 1.4.6) function of Rsubread (Liao et al, 2014) and analyzed by DESeq2 version 1.12.4 (Love et al, 2014). As each lot of primary MEFs were traceable with or without 4OHT treatment, pairwise comparisons were performed in each individual MEF for evaluating DE with or without the effects of cre-induced recombination.

## Pathway analysis

We used the DE genes identified in RNA-Seq analysis above (FDR < 0.05) using Ingenuity Pathway Analysis (QIAGEN) or KEGG. KEGG annotations were added using the *R* package clusterProfiler (Yu et al, 2012). Mapped KEGG enrichments were visualized using FuncTree (Uchiyama et al, 2015) available at https://bioviz.tokyo/functree/.

## Metabolome analysis

Cellular metabolites were analyzed by CE-TOFMS using primary MEFs for *Crk* or *Crkl* single gene deficiency and *Crk/Crkl* double deficiency as well as their uninduced controls as previously described (Uetaki et al, 2015). Results obtained from three independent samples were compared for each genotype between deficiency-induced and uninduced MEF groups using Welch's *t* test for each metabolite ($P < 0.05$).

## ChIP and ChIP-Seq

MEFs induced *Crk/Crkl* deficiency were harvested 30 h after removal of 4OHT, along with control MEFs treated with vehicle instead of 4OHT. Samples were prepared using SimpleChIP Plus Enzymatic Chromatin IP Kit (#9005; Cell Signaling Technology) as recommended in the manufacturer's protocol. Antibodies used were anti-histone H3 acetylated lysine 27 rabbit polyclonal antibody (ab4729; Abcam) and anti-RNA polymerase 2 CTD repeat YSPTSPS (phospho-S5) mouse monoclonal antibody (clone 4H8, ab5408; Abcam). For immunoprecipitation, Dynabeads protein G or M-280 sheep anti-mouse IgG (Thermo Fisher Scientific) was used to best match the species range and specificity for each primary antibody. For quantitative analysis of selected glycolysis genes, Pol2 ChIP samples were used for real-time PCR using SYBR Green with the primer pairs listed on Table S3. For ChIP-Seq experiments, high-throughput sequencing was conducted in an Illumina HiSeq 2500 for 36-bp single end reads. All ChIP-Seq data were first processed with Cutadapt and FastQC under the wrapper software Trim Galore v0.4.4 with the "-q 30" option in Cutadapt to trim off low quality ends (https://www.bioinformatics.babraham.ac.uk/projects/trim_galore/). The sequence output was then aligned to the mouse reference genome GRCm38.p3 using Bowtie v1.1.2 with the "-m 1" option (Langmead et al, 2009). Duplicates were removed from aligned reads using PICARD v1.14 (https://broadinstitute.github.io/picard/). The mapped reads were then standardized for each experiment to an effective mouse genome size of 2,652,783,500 bases as "reads per genome coverage or content" (RPGC) using the utility package deepTools v2.1.1 (Ramírez et al, 2016). Utilities in deepTools were also used for downstream analysis of normalized ChIP-Seq results. ChIP-Seq results were normalized against the background signals obtained from whole cell extracts for corresponding cell groups.

## Automated image quantification

Image segmentation was performed using a custom MATLAB script (available upon request; MATLAB is a programming language available from Mathworks). First, we acquired a set of images of CellMask, DAPI, and anti-vinculin (or anti-FAK) staining to segment the cells, nuclei, and focal adhesions, respectively, under a 40× objective lens equipped in IN Cell Analyzer 2500HS (GE Healthcare). We used a previously published method based on phase stretch transform to segment nuclei and focal adhesions (Asghari & Jalali, 2015), while performing empirical optimizations of the input parameters. To segment the cells, we first smoothed the images and used a threshold based on the average intensity of the dimmest

20% of pixels in the CellMask channel corresponding to background pixels. We used the DAPI staining to determine if any segmented regions contained multiple nuclei corresponding to under-segmented regions. These regions were re-segmented with a higher threshold and expanded by region growing. The intersection of these expanded regions was used to segment this larger region into single cells. After this step, we removed any remaining regions with 0 or multiple nuclei.

We noticed that blebbing cells consistently showed rough cell boundaries having bead-like bulging membrane protrusions with high curvatures, small cell/nuclear area ratio, and a high cytoplasmic intensity of vinculin signals relative to that of the nucleus. To estimate the boundary curvature for each single cell, we first performed smoothing edge boundaries by Savitzky–Golay filter (Diederick, 2019). An instantaneous curvature was estimated for each set of neighboring points using a code deposited at the MATLAB Central File Exchange (Mjaavatten, 2019), where the curvature is defined as $1/r_i$ ($r_i$ is the radius for a point, $P_i$). To standardize threshold parameters, we analyzed five randomly selected images from each group (2-by-2 groups: with or without 3 mM 2DG× with or without *Crk/Crkl* deficiency induction) and manually identified blebbing and non-blebbing cells among the segmented cells (Fig S9). In automated analysis, the cells were then ruled "blebbing" when they have a combination of the three standardized threshold parameters: an average single-cell curvature larger than 0.029 pixel$^{-1}$, a ratio of nuclear to cytoplasmic vinculin intensity <1.15, and a cell/nuclear area size ratio <4.5 (Fig S9). Using these standardized parameters, "blebbing" cells were then identified in a total of 7,409 MEFs segmented (Fig 6B). As some cells appear to be outliers (blue arrows in Fig S9B), we filtered out the cells having the DAPI-positive area size smaller than 1,000 pixels or with the cell area greater than 150,000 pixels, whereas the median nuclear and cell area sizes were 8,638 and 35,665 pixels, respectively. The filtering process removed 809 cells and 148 cells from analysis, respectively.

For quantification of focal adhesions, we used the cell and nuclear segmentations to remove noise from our focal adhesion segmentation by ruling out segmentation outside the cells or inside the nucleus. We further refined the focal adhesion selection by removing areas smaller than 30 pixels or lower intensity than the cell average staining intensity. We confirmed that differences in the number of focal adhesions segmented in each condition could not be attributed to systematic differences in rates of segmentation errors for different image sets (Figs 7G and S11).

### Resources

The knockout-ready *Crk* conditional strain will be available through the Jackson Laboratory (JAX Stock #032874). The RNA-Seq and ChIP-Seq data have been deposited to the DDBJ (www.ddbj.nig.ac.jp) and have been assigned the accession numbers DRA007302 and DRA007305, respectively. The deposited read data will be available via the BioProject page at NCBI as the BioProjects PRJDB7421 and PRJDB7413, respectively (www.ncbi.nlm.nih.gov/bioproject/).

## Supplementary Information

## Acknowledgements

The authors thank VE Papaioannou for the *Tbx1* null strain, P Soriano for the *Meox2$^{cre}$* and *R26$^{FLPeR}$* strains, Y Saga for the *Mesp1$^{cre}$* strain, M Matsuda for the C3G-F plasmid, L Degenstein and The Transgenic and ES Cell Technology Core for assisting generation of the *Crk* conditional mutant strain. This work was supported in part by research grants from JSPS (17H06299, 17H06302, and 18H04031), the Nagase Science Technology Foundation, and Astellas Foundation for Research on Metabolic Disorders to M Okada; from JSPS (17H06299) to Y Suzuki, from JST PRESTO (JPMJPR1507) and Japan AMED (17ek0109187h0002) to T Yamada; and from JSPS (15H01522, 16H04901, 17H05654, and 18H04805) and JST PRESTO (JPMJPR1537) to S Fukuda.

### Author Contributions

A Imamoto: conceptualization, data curation, formal analysis, supervision, investigation, and writing—original draft, review, and editing.
S Ki: investigation.
L Li: investigation.
K Iwamoto: data curation and formal analysis.
V Maruthamuthu: investigation and methodology.
J Devany: software, investigation, and methodology.
O Lu: investigation.
T Kanazawa: investigation.
S Zhang: investigation.
T Yamada: data curation, software, formal analysis, and visualization.
A Hirayama: investigation.
S Fukuda: investigation and methodology.
Y Suzuki: investigation and methodology.
M Okada: conceptualization, supervision, funding acquisition, and writing—review and editing.

### Conflict of Interest Statement

The authors declare that they have no conflict of interest.

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
