## [Reviewer comments · Life Science Alliance]

Life Science Alliance

Essential role of the Crk family-dosage in DiGeorge-like anomaly and metabolic homeostasis

Akira Imamoto, Sewon Ki, Leiming Li, Kazunari Iwamoto, Venkat Maruthamuthu, John Devany, Ocean Lu, Tomomi Kanazawa, Suxiang Zhang, Takuji Yamada, Akiyoshi Hirayama, Shinji Fukuda, Yutaka Suzuki, and Mariko Okada-Hatakeyama

DOI: <https://doi.org/10.26508/lsa.201900635>

Corresponding author(s): Akira Imamoto, The University of Chicago and Mariko Okada-Hatakeyama, Osaka University

Review Timeline:

Submission Date:	2019-12-23
Editorial Decision:	2019-12-31
Revision Received:	2020-01-14
Editorial Decision:	2020-01-16
Revision Received:	2020-01-20
Accepted:	2020-01-21

Scientific Editor: Andrea Leibfried

Transaction Report:

. Please note that the manuscript was previously reviewed at another journal and the reports were taken into account in inviting a revision for publication at Life Science Alliance prior to submission to Life Science Alliance:

Referee #1 Review

Report for Author:

In their paper "Essential role of the Crk family-dosage in DiGeorge-like anomaly and metabolic homeostasis" the authors first analyse the effect of the conditional deletion of Crk (and Crkl) in mouse embryos and later in mouse embryo fibroblasts. The key embryonic phenotype upon germline deletion of Crk is described as phenocopying part of DiGeorge syndrome. In addition, they report the embryonic effect of mesoderm-specific deletion of Crk and/or Crkl. While the full deletion of both genes leads to a very severe gastrulation stage phenotype with general developmental delay, mesoderm specific deletion of all but one copy of Crk and Crkl leads to heart anomalies, smaller somites and enlarged presomitic mesoderm. The authors also demonstrate a genetic

interaction between Crk and Tbx1, a gene previously associated with DiGeorge syndrome.

The majority of results presented in this manuscript are collected, however, in MEFs, which show a cell spreading and cell size phenotype when Crk/Crkl are deleted. The transcriptomic analysis in Crk/Crkl deficient MEFs showed a downregulation of glycolytic genes. The metabolic effect is analyzed using metabolomics and the authors provide evidence that it is mediated via reduced Hif1a production. Finally, they show, in MEFs, that they can rescue these effects with an overexpression of the guanine-nucleotide exchange factor for the small Gprotein Rap1, Rapgef1, a known Crk interacting protein.

This study contains a wealth of molecular data and overall, appears to be done very carefully and systematic.

The key (obvious) criticism is that most of the analysis and molecular and metabolic findings are made using MEFs. Importantly, the functional connection of the molecular findings in MEFs, including the metabolic alterations in MEFs, to the embryonic phenotype remain unaddressed, which to me is the key factor lowering the impact of this study. Along this line, it needs to be noted that metabolic activities are highly regulated, in time and space, in embryos during and after gastrulation. It is hence unclear, whether MEFs do reflect any of these tissue specific metabolic activity patterns.

Here are further issues listed:

- 1) The embryonic phenotype in Crk/Crkl mutants with mesoderm specific deletion is only very rudimentary described. A more systematic analysis of phenotype, also at molecular and metabolic level, in embryos is needed in my view. This analysis does not need to be as comprehensive as done in MEFs, of course, but needs to test key links, i.e. are metabolic genes affected and glycolysis perturbed in embryos?
- 2) Is it possible to affect/modulate the phenotypic outcome by perturbation of metabolism, for instance by culturing embryos in a range of glucose concentration? This could be a highly relevant avenue, as it also may relate to the findings that in DiGeorge syndrome, the phenotypic manifestation is highly variable, even among affected family members. Hence a modulating role of metabolic activities is possible.
- 3) While ideally done in embryos, also role of metabolism in causing the cellular phenotype in MEFs needs to be tested, functionally. Does alteration of metabolism cause the cellular phenotype observed? For instance, does inhibition of glycolysis phenocopy the cellular findings?
- 4) Vice versa: are any of the cellular effects found in MEFs visible in embryos?
- 5) Is the rescue using Rapgef1 occurring also in vivo?
- 6) Minor: How is the metabolite data in Fig. 4A normalized to take altered cell size into account?

Referee #2 Review

Report for Author:

This is an articulated manuscript with some sophisticated mouse genetics and extensive cell biology experiments that shed some light on the developmental functions of the Crk "family" and adds a novel metabolic insight into the function of these genes.

The manuscript does not address issues related to DiGeorge syndrome, therefore the title and frequent references to the human disease throughout the manuscript are inappropriate and should be substantially toned down, perhaps limited to a paragraph in the discussion.

My major criticism is that many of the findings (and conclusions) of the manuscript are based on a cell culture system and not validated in vivo in a developing embryo. In addition, the cell type used

for the tissue culture experiments (MEF) may not be the major target of CRKs in vivo. I think that it is essential to confirm in vivo at least the most critical results obtained in a tissue culture dish.

Major points

-Description of the phenotype. The embryological phenotype of mutants is described in a superficial manner. There is no indication as to how many embryos were analysed per genotype, and how. There is no evaluation of penetrance and expressivity of the defects (with the exception of Tbx1, which only refers to the Tbx1-Crk interaction experiment).

I have doubts about the authors' definition of thymic hypoplasia (see also my comments about Fig. S3).

In addition, the authors make statements such as "It is also noteworthy that embryos with only one copy of either Crkl or Crk (Crk f/f;Crkl f2/+;Mesp1Cre/+ or Crk f/+;Crkl f2/f2;Mesp1Cre/+, respectively) showed virtually identical phenotypes" but I see no detailed analyses of these phenotypes. How can the authors claim that are "virtually identical"? What phenotypic features do they refer to?

- Cell phenotype. Most of the manuscript concerns tissue culture experiments. While the results obtained are interesting, I have little enthusiasm for the data, as they are not validated in any way in a developmental context. Therefore, their significance and importance for the developmental defects of mutants remains speculative. The authors should validate at least some of their cell culture experiments in vivo.

- The authors state: "The results above suggest that mesodermal cells may provide a useful system to investigate the shared functions of Crk and Crkl". However, Crk f/f;Mesp1Cre/+, Crkl f2/f2;Mesp1Cre/+, and Crk f/+;Crkl f2/+;Mesp1Cre/+ are said to have no phenotypic anomalies, in contrast to individual germ line KOs or double hets. Thus, deletion in the anterior mesoderm is not sufficient to recapitulate the KOs, suggesting that other tissues, such as endoderm, ectoderm or neural crest may be important targets of Crk/Crkl function.

- It seems that some of the cell biology tests performed here were also performed and published using Tbx1 mutants (<https://doi.org/10.1093/hmg/ddz058>). Perhaps the authors should discuss the published results as they may help the interpretation of the Tbx1-Crk interaction phenotype.

Minor

- The title should be modified because the article does not address anything concerning the human disease. In addition, the term "DiGeorge-like anomaly" is meaningless.

- Analogously, the sentence in the Abstract " Here we show that a 50% reduction of the family-combined dosage phenocopies DiGeorge/del22q11 syndrome in mice" is simply untrue and should be removed. I do not know any animal model that is really a "phenocopy" the 22q11.2DS phenotype.

- DiGeorge-like anomaly, DiGeorge syndrome, are used freely and mostly inappropriately throughout the manuscript. In fact, the relationship between the 22q11.2DS and Crk is totally unproven, while the relationship with Crkl is at the moment limited to kidney defects. Indeed, there is no evidence indicating a contribution of the Crkl gene to the cardiac phenotype because patients with the large deletion (including Crkl) or with a smaller deletion (not including Crkl) have the same cardiac phenotype.

TGA is very rare in 22q11.2DS, certainly not a typical defect in this syndrome.

- Fig. 1, panels F and G have no controls of normal anatomy.

- Figure S3B, embryo#1 is said to have "abnormal origin of the RS" but I do not see it. The panel is also inadequate to show IAAB. The picture is confusing and it seems that there is some tissue covering the aortic arch. The thymic lobi of the embryo #4 appear within the normal size range, why are they defined as "hypoplastic"?

- The authors should specify the statistical methods used to quantify the cell biology experiments, how many times they have repeated the experiments, etc.

December 31, 2019

Re: Life Science Alliance manuscript #LSA-2019-00635-T

Prof. Akira Imamoto
The University of Chicago
Ben May Department for Cancer Research
929 E. 57th Street, GCIS-W332
Chicago, IL 60637

Dear Dr. Imamoto,

Thank you for transferring your manuscript entitled "Essential role of the Crk family-dosage in DiGeorge-like anomaly and metabolic homeostasis" to Life Science Alliance. The manuscript was assessed by expert reviewers at another journal before, and the editors transferred those reports to us with your permission.

The reviewers appreciated the quality of the data provided, but thought that the in vivo relevance of your findings remains rather unclear. You used primary MEFs for your assays and we concluded that the major concern raised by the reviewers does not preclude publication in Life Science Alliance. We would thus like to invite you to submit a revised version of your manuscript to us. We would expect a full point-by-point response and accordingly text changes. Points 1 and 3 of reviewer #1 should get addressed experimentally. The phenotype description should also get improved (reviewer #2), controls added (minor point 4 of reviewer #2) and reviewer #2's comment regarding statistics and replicates should get addressed.

Thank you for this interesting contribution to Life Science Alliance. We are looking forward to receiving your revised manuscript.

Sincerely,

Andrea Leibfried, PhD
Executive Editor
Life Science Alliance
Meyerhofstr. 1

69117 Heidelberg, Germany
t +49 6221 8891 502
e a.leibfried@life-science-alliance.org
www.life-science-alliance.org

B. MANUSCRIPT ORGANIZATION AND FORMATTING:

Point-to-point responses to reviewer critiques

REVIEWER 1

1. *The embryonic phenotype in Crk/Crkl mutants with mesoderm specific deletion is only very rudimentary described. A more systematic analysis of phenotype, also at molecular and metabolic level, in embryos is needed in my view. This analysis does not need to be as comprehensive as done in MEFs, of course, but needs to test key links, i.e. are metabolic genes affected and glycolysis perturbed in embryos?*

While we agree that it would be of significance to demonstrate abnormal metabolic gene expression in mutant embryos, we believe that such experiments will be beyond the scope of the current study based on the following reasons. Individual *Crk* or *Crkl* homozygous deficiency as well as compound heterozygosity is embryonic lethal by late gestation. Further gene dosage reduction in either *Crk* or *Crkl* then leads to an early gastrulation phenotype even when limited to the mesoderm with poor recovery rates. Therefore, while undoubtedly useful, evaluations of metabolic genes as suggested would not be straightforward due partly to their complex cross designs (highly labor-intensive). In addition, anticipated morphological defects are also secondary or tertiary to a primary defect in vivo, while there may likely be dosage-effects. Therefore, careful designs and interpretations will be needed for such in vivo analyses.

We would also like to point out that we have chosen to isolate primary MEFs from phenotypically normal embryos before induction of *Crk/Crkl* deficiency to better control experimental conditions. 'Primary' cells retain conditions much closer to that of embryos than established/immortalized MEF lines. We emphasize that our current study goal was to generate novel hypotheses so that we and others can design appropriate experiments in future. Hence, a simpler well-controlled model was preferred than complex in vivo models for an initial study.

2. *Is it possible to affect/modulate the phenotypic outcome by perturbation of metabolism, for instance by culturing embryos in a range of glucose concentration? This could be a highly relevant avenue, as it also may relate to the findings that in DiGeorge syndrome, the phenotypic manifestation is highly variable, even among affected family members. Hence a modulating role of metabolic activities is possible.*

This is related to the second question above. We believe that this question would be more appropriately addressed by using mouse models in vivo, by not only *Crk/Crkl* but also including *Tbx1* mutant models. Nevertheless, in the spirit of the proposed in vivo experiment, we carried out a new experiment in MEFs with 2-deoxy-D-glucose (new Figures 6B and S9) – please see our response to Point 3 below.

3. *While ideally done in embryos, also role of metabolism in causing the cellular phenotype in MEFs needs to be tested, functionally. Does alteration of metabolism cause the cellular phenotype observed? For instance, does inhibition of glycolysis phenocopy the cellular findings?*

As this point was also raised by the editor, we have added new experiments in which we have restricted glucose metabolism by addition of 2-deoxy-D-glucose (2DG). We have found that 2DG causes cell blebbing in MEFs. *Crk/Crkl* deficiency-induced MEFs show higher frequency of blebbing than control MEFs, thus suggesting that *Crk/Crkl* deficiency makes MEFs more sensitive to glucose availability.

The new experiments conducted above did not yield a clear mechanism that links focal adhesions and glucose metabolism. However, the results suggest that regulation of glycolysis lies downstream of cell-matrix adhesions, since activated Rapgef1 (C3G) could rescue both focal adhesions and metabolism (Figure 7). It is also widely known that restricting cell-matrix adhesions affects cell survival in many normal adherent cell types. We have included this interpretation in Discussion in our revised manuscript.

4. *Vice versa: are any of the cellular effects found in MEFs visible in embryos?*

In order to be able to observe the cellular effects *in vivo*, we will have to design new studies to make it possible. We do observe a low cellularity in affected tissues *in vivo* – which will need to be evaluated in more systematic and objective methods.

5. *Is the rescue using Rapgef1 occurring also in vivo?*

Although we also agree that it would be an important experiment, such experiments will require a new set of mouse models to be able to set up a feasible system (assuming a dominant effect of activated Rapgef1 if expressed without a conditional approach). Therefore, once again, it is beyond the scope of the current study.

6. *Minor: How is the metabolite data in Fig. 4A normalized to take altered cell size into account?*

The metabolite data were normalized by cell numbers which was calculated by DNA contents. We have added this information in the legend.

REVIEWER 2

1. *- Description of the phenotype. The embryological phenotype of mutants is described in a superficial manner. There is no indication as to how many embryos were analysed per genotype, and how. There is no evaluation of penetrance and expressivity of the defects (with the exception of TabS1, which only refers to the Tbx1-Crk interaction experiment). I have doubts about the authors' definition of thymic hypoplasia (see also my comments about Fig. S3).*

To clarify the description, we added new figures. Please see the new controls added in Figure 1. In addition, we have added additional case for the *Crk/Crkl* compound heterozygous phenotype as a piece of evidence for high penetrance (and reproducibility). The text is revised to describe common phenotypic aspects.

We regret that our data did not convince the reviewer regarding our categorical judgements of thymic size. However, it should be clear to the reviewer that we do have athymic cases in addition to hypoplasia cases. In addition, our categorical judgements are based on years of mouse work to analyze mouse phenotypes including thymic defects as we published in several papers. For example, in the 2006 paper (Guris et al., Dev Cell 2006), we measured the thymic size to know normal vs abnormal (hypoplastic) thymic lobes in mouse embryos at E16.5. Therefore, we believe that our expertise is sufficient for calling thymic hypoplasia beyond the normal range that we have observed in numerous littermates at E16.5. Furthermore, we have made it clear that several abnormal/hypoplastic cases have ectopic/cervical lobes in the revised manuscript (as shown in Figure 1B).

2. *In addition, the authors make statements such as "It is also noteworthy that embryos with only one copy of either Crkl or Crk (Crk f/f;Crklf2/+;Mesp1Cre/+ or Crk f/+;Crkl f2/f2;Mesp1Cre/+, respectively) showed virtually identical phenotypes" but I see no detailed analyses of these phenotypes. How can the authors claim that are "virtually identical"? What phenotypic features do they refer to?*

We have rephrased from “virtually identical” to “similar”. However, both phenotypes include a few mesodermal defects such as delayed somitogenesis and development of heart chambers (including abnormal heart tube looping), as clearly described in the original manuscript.

3. *- Cell phenotype. Most of the manuscript concerns tissue culture experiments. While the results obtained are interesting, I have little enthusiasm for the data, as they are not validated in any way in a developmental context. Therefore, their significance and importance for the developmental defects of mutants remains speculative. The authors should validate at least some of their cell culture experiments in vivo.*

As indicated in our response to Reviewer 1, while we recognize importance of in vivo experiments, they are beyond the scope of the current study.

4. *- The authors state: "The results above suggest that mesodermal cells may provide a useful system to investigate the shared functions of Crk and Crkl". However, Crk f/f;Mesp1Cre/+, Crkl f2/f2;Mesp1Cre/+, and Crk f/+;Crkl f2/+;Mesp1Cre/+ are said to have no phenotypic anomalies, in contrast to individual germ line KOs or double hets. Thus, deletion in the anterior mesoderm is not sufficient to recapitulate the KOs, suggesting that other tissues, such as endoderm, ectoderm or neural crest may be important targets of Crk/Crkl function.*

The phenotypes resulted from global deficiency are likely a synthesis from defects in multiple tissue types that likely have functional interactions. As shown in the in vivo results, this view doesn't conflict with our statement that *Crk* and *Crkl* are indeed required for early mesoderm development. Nevertheless, we have added a few sentences to clarify our interpretations.

5. - *It seems that some of the cell biology tests performed here were also performed and published using Tbx1 mutants (<https://doi.org/10.1093/hmg/ddz058>). Perhaps the authors should discuss the published results as they may help the interpretation of the Tbx1-Crk interaction phenotype.*

The *Tbx1* paper was not available at the time of our initial submission of the manuscript to the previous Journal. We thank the reviewer to point out similarities that we have also indicated in Discussion of the original manuscript (although we did not cite the paper). In our revised manuscript, we have cited the *Tbx1* paper as it does strengthen our original hypothesis in Discussion.

Minor

1. - *The title should be modified because the article does not address anything concerning the human disease. In addition, the term "DiGeorge-like anomaly" is meaningless.*

We respectfully disagree with this comment. No single set of clinical definitions thus far perfectly agree with the molecular definitions due to high degrees of clinical variations in humans (please see our response below for Point 3). Previous studies have failed to establish firm genotype-phenotype correlations for various sizes and positions of deletions found in 22q11.2DS patients. Having a few clinical features without being a phenocopy are generally enough to raise a suspicion of 22q11.2DS, which would be confirmed by molecular diagnosis. This is the fact described in many clinical and research articles in the field despite this assertion from the reviewer. Furthermore, we did not claim that *Crk* is a DiGeorge candidate gene, while we describe experimental observations in genetically modified mice and cells.

2. - *Analogously, the sentence in the Abstract " Here we show that a 50% reduction of the family-combined dosage phenocopies DiGeorge/del22q11 syndrome in mice" is simply untrue and should be removed. I do not know any animal model that is really a "phenocopy" the 22q11.2DS phenotype.*

This comment is debatable as compound heterozygosity of *Crk* and *Crkl* clearly has a lethal phenotype that includes a few aspects of 22q11.2DS. We should keep it in mind that all aspects of "DiGeorge spectrum" have partial penetrance in patients and animal models. No single individual exhibits all spectrum in human cases. Instead, we believe that the reviewer's point appears to boil down to the word 'phenocopy'. We have rephrased the sentence by replacing the word 'phenocopy' to objectively state phenotypic similarities.

3. - *DiGeorge-like anomaly, DiGeorge syndrome, are used freely and mostly inappropriately throughout the manuscript. In fact, the relationship between the 22q11.2DS and Crk is totally unproven, while the relationship with Crkl is at the moment limited to kidney defects. Indeed, there is no evidence indicating a contribution of the Crkl gene to the cardiac phenotype because patients with the large deletion (including Crkl) or with a smaller deletion (not including Crkl) have the same cardiac phenotype. TGA is very rare in 22q11.2DS, certainly not a typical defect in this syndrome.*

We are simply perplexed by this comment. We clearly stated in the original submission that there is no clinical/genetic evidence to link *CRK* to 22q11.2DS, besides the fact that *CRK* is not a 22q11 gene. Further, we did not hypothesize that *CRK* might be a DiGeorge candidate gene anyway. Instead, this manuscript presents some evidence that organogenesis affected in 22q11.2DS or related syndrome may rely on pathways or mechanisms shared by *CRK* and *CRKL* (and *TBX1*).

Moreover, it appears that the reviewer misunderstood the ramifications of the two *CRKL* kidney defect papers published in NEJM and PNAS in 2018. The patient cohort was selected from GU patient pools in the NEJM paper, not from 22q11.2DS patient pools. The results reported in NEJM and PNAS should not be interpreted as if *CRKL* were important ONLY for kidney development. Rather, these published papers support the possibility that haploinsufficiency of *CRKL* (clearly not *TBX1*) contributes to the GU defects manifested in a significant subset of 22q11.2DS patients, as *Crkl* haploinsufficiency also generated GU defects albeit in partial penetrance in mice.

On the same token, we were also mystified by the reviewer's statement that "*Indeed, there is no evidence indicating a contribution of the Crkl gene to the cardiac phenotype because patients with the large deletion (including Crkl) or with a smaller deletion (not including Crkl) have the same cardiac phenotype*". This statement is inaccurate. The well-established notion in the field is that the hemizyosity of the 22q11 region alone does not explain highly variable expressivity and penetrance (reviewed by Robin, N.H., and Shprintzen, R.J. 2005. "Defining the Clinical Spectrum of Deletion 22q11.2". J. Pediatr. 147, 90–96; McDonald-McGinn, et al. 2015. "22q11.2 deletion syndrome". Nat. Rev. Dis. Prim. 1, 15071.). Even the same deletion within a family could be associated with variable outcome. DiGeorge-like phenotypes have been reported in non-overlapping deletions in 22q11 (e.g., proximal vs distal deletions). Several distal deletions including *CRKL* but not *TBX1* have been reported among 22q11.2DS patients. These results have failed the single-gene-etiology hypothesis.

A recent report presents significant association of outflow tract defects with non-coding SNPs that are predicted to downregulate nearby *CRKL* in the hemizygous 22q11.21 chromosomal region in a large cohort of 22q11.2DS patients (Zhao, et al. 2020. "Complete Sequence of the 22q11.2 Allele in 1,053 Subjects with 22q11.2 Deletion Syndrome Reveals Modifiers of Conotruncal Heart Defects". Am. J. Hum. Genet. 106, 26–40.). Thus, a reduced dosage of *CRKL* (less than 50% of normal level) likely contributes to the phenotypic outcome as an important modifier. We have included this information in the revised manuscript. In addition, mouse studies (which others and we conducted independently) have shown that *CRKL* is an important gene for cardiovascular development as a dosage sensitive gene since the mouse phenotypes have similarities to 22q11.2DS patients.

In the reviewer's statement, it is true that TGA is rare in 22q11.2DS, whereas thymic defects and abnormal patterns of great arteries we found in our animal models are common in 22q11.2DS. It should be kept in mind that TGA is a severe form of conotruncal malformation associated with abnormal alignment, which is affected in 22q11.2DS patients.

Although the reviewer listed Point 3 as a minor point, we fear that the reviewer's overall assessment might have been biased based on the reviewer's limited knowledge on the previous studies of this syndrome in patients and in animal models.

4. - *Fig. 1, panels F and G have no controls of normal anatomy.*

We have enhanced Figure 1 by including normal controls where possible (new panels A'-D'). Regrettably, previous panels F and G (new panels G and H) were only decent images taken at the time after intracardiac ink injection. However, normal anatomy is well established in the research field at E16.5 in mice. To strengthen our case, we have included another case of an interrupted arch of aorta type B recovered in the same litter in additional panel in (new panel I). Our laboratory has extensive experience in analyzing arch artery/great artery/heart phenotypes in mid-late gestation in the past as described above.

5. - *Figure S3B, embryo#1 is said to have "abnormal origin of the RS" but I do not see it. The panel is also inadequate to show IAAB. The picture is confusing and it seems that there is some tissue covering the aortic arch. The thymic lobi of the embryo #4 appear within the normal size range, why are they defined as "hypoplastic"?*

Normal anatomy of the arch arteries is well established in mice. RSA (right subclavian artery), for example, has a specific position at which it bifurcates from right common carotid artery about their junction to the arch of aorta. If there was no normal RSA bifurcation, it is clearly abnormal (hence, 'abnormal origin'). This should be obvious from the image shown. In addition, as stated in Point 4 above, newly added case also strengthens relative abundance of IAAB cases. The size of thymic 'lobes' varies as the reviewer pointed out, while there is a normal range of variations. As shown in newly added controls, the degrees of hypoplastic and/or ectopic lobes would be obvious in Figure 1.

6. - *The authors should specify the statistical methods used to quantify the cell biology experiments, how many times they have repeated the experiments, etc.*

Most cell biology experiments have had statistical evaluations clearly described (Figs 2, 7). Perhaps one exception was cell staining in Figure 7F. While reduced focal adhesions were noted in many staining experiments, in our revision, we have provided an unbiased evaluation using an automated system using a custom code to evaluate a large number of cells (thousands), beyond one's impression or manual counts from a few fields of microscopic images. New data are presented in Figures 7G and S11.

Responses to the Editor's comments

Points 1 and 3 of reviewer #1 should get addressed experimentally. The phenotype description should also get improved (reviewer #2), controls added (minor point 4 of reviewer #2) and reviewer #2's comment regarding statistics and replicates should get addressed.

Reviewer 1's Point #1: We have provided our justification of our use of "primary" MEFs for large part of our study reported in the manuscript, as it may have been obscure in the original submission.

Reviewer 1's Point #3. We have added a new experiment as shown in Figure 6B using 2-deoxy-D-glucose. Although we initially conducted an experiment to rescue *Crk/Crkl* deficiency by introducing membrane-anchored Akt, as have done with membrane-anchored Rapgef1, the activated Akt induced a highly transformed cell phenotype in wild type MEFs. Thus, we concluded it wasn't suited for morphological evaluations, therefore not included in the revised manuscript. Instead, we took on your advice to use 2DG to confirm a link between glucose metabolism and *Crk/Crkl* deficiency (Figures 6B and S9).

Reviewer 2's comment on the mouse phenotypes including Minor Point #4 (which is noted as Minor Point #6 in our point-to-point responses): We have added controls as well as additional panels in Figure 1 and enhanced phenotypic descriptions. With regard to the mesoderm-specific gene disruptions, we feel that the original manuscript already had adequate description. We interpreted that the comment about phenotypes was made because Reviewer 2 had an issue with the use of the term 'phenocopy'. Thus, we removed the term and rephrased the corresponding sentences as you will find in our responses to Reviewer 2. In addition, we have conducted additional unbiased experiments to quantify focal adhesions and added new results in Figures 7G and S11 by incorporating automated analysis of cell morphology over thousands of cells.

We believe that the revised manuscript has been strengthened significantly. We thank the editor for her constructive comments and guidance.

January 16, 2020

RE: Life Science Alliance Manuscript #LSA-2019-00635-TR

Prof. Akira Imamoto
The University of Chicago
Ben May Department for Cancer Research
929 E. 57th Street, GCIS-W306
Chicago, IL 60637

Dear Dr. Imamoto,

Thank you for submitting your revised manuscript entitled "Essential role of the Crk family-dosage in DiGeorge-like anomaly and metabolic homeostasis". I appreciate the introduced changes and would thus be happy to publish your paper in Life Science Alliance pending final revisions necessary to meet our formatting guidelines:

- Please add a callout in the manuscript text to figure S8
- Please upload all supplementary figures as individual files; the supplementary tables and supplementary figure legends can go into the main manuscript file.
- Please add the legend to the three datasets in the excel spreadsheets themselves
- Please move the following statements into the Methods section: "The RNA-Seq and ChIP-Seq data have been deposited to the DDBJ (www.ddbj.nig.ac.jp) and have been assigned the accession numbers DRA007302 and DRA007305, respectively. The deposited read data will be available via the BioProject page at NCBI (www.ncbi.nlm.nih.gov/bioproject/). The knockout-ready Crk conditional strain will be available through the Jackson Laboratory (JAX Stock #032874)."

A. FINAL FILES:

-- High-resolution figure, supplementary figure and video files uploaded as individual files: See our detailed guidelines for preparing your production-ready images, <http://www.life-science->

alliance.org/authors

B. MANUSCRIPT ORGANIZATION AND FORMATTING:

Sincerely,

Andrea Leibfried, PhD
Executive Editor
Life Science Alliance
Meyerhofstr. 1
69117 Heidelberg, Germany
t +49 6221 8891 502
e a.leibfried@life-science-alliance.org

January 21, 2020

RE: Life Science Alliance Manuscript #LSA-2019-00635-TRR

Prof. Akira Imamoto
The University of Chicago
Ben May Department for Cancer Research
929 E. 57th Street, GCIS-W306
Chicago, IL 60637

Dear Dr. Imamoto,

Thank you for submitting your Research Article entitled "Essential role of the Crk family-dosage in DiGeorge-like anomaly and metabolic homeostasis". It is a pleasure to let you know that your manuscript is now accepted for publication in Life Science Alliance. Congratulations on this interesting work.

DISTRIBUTION OF MATERIALS:

Again, congratulations on a very nice paper. I hope you found the review process to be constructive and are pleased with how the manuscript was handled editorially. We look forward to future exciting submissions from your lab.

Sincerely,
